# Berberine is an insulin secretagogue targeting the KCNH6 potassium channel

Miao-Miao Zhao [1,2], Jing Lu[1,2], Sen Li[2], Hao Wang[1,2,3], Xi Cao[1,2], Qi Li[2], Ting-Ting Shi[1,2], Kohichi Matsunaga[3], Chen Chen [4], Haixia Huang[5], Tetsuro Izumi [3] & Jin-Kui Yang [1,2 ✉]

*Coptis chinensis* is an ancient Chinese herb treating diabetes in China for thousands of years. However, its underlying mechanism remains poorly understood. Here, we report the effects of its main active component, berberine (BBR), on stimulating insulin secretion. In mice with hyperglycemia induced by a high-fat diet, BBR significantly increases insulin secretion and reduced blood glucose levels. However, in mice with hyperglycemia induced by global or pancreatic islet β-cell-specific *Kcnh6* knockout, BBR does not exert beneficial effects. BBR directly binds KCNH6 potassium channels, significantly accelerates channel closure, and subsequently reduces KCNH6 currents. Consequently, blocking KCNH6 currents prolongs high glucose-dependent cell membrane depolarization and increases insulin secretion. Finally, to assess the effect of BBR on insulin secretion in humans, a randomized, double-blind, placebo-controlled, two-period crossover, single-dose, phase 1 clinical trial (NCT03972215) including 15 healthy men receiving a 160-min hyperglycemic clamp experiment is performed. The pre-specified primary outcomes are assessment of the differences of serum insulin and C-peptide levels between BBR and placebo treatment groups during the hyperglycemic clamp study. BBR significantly promotes insulin secretion under hyperglycemic state comparing with placebo treatment, while does not affect basal insulin secretion in humans. All subjects tolerate BBR well, and we observe no side effects in the 14-day follow up period. In this study, we identify BBR as a glucose-dependent insulin secretagogue for treating diabetes without causing hypoglycemia that targets KCNH6 channels.

[1] Department of Endocrinology, Beijing Tongren Hospital, Capital Medical University, 100730 Beijing, China. [2] Beijing Key Laboratory of Diabetes Research and Care, Beijing Diabetes Institute, 100730 Beijing, China. [3] Laboratory of Molecular Endocrinology and Metabolism, Department of Molecular Medicine, Institute for Molecular and Cellular Regulation, Gunma University, Maebashi, Japan. [4] School of Biomedical Sciences, University of Queensland, Brisbane, QLD 4072, Australia. [5] Department of Physiology and Pathophysiology, School of Basic Medical Sciences, Capital Medical University, 100069 Beijing, China. ✉email: jkyang@ccmu.edu.cn

Natural plants are rich sources of novel drugs. During the past three decades, ~65% of all small-molecule drugs approved for marketing have been traced to or were inspired by natural plant extracts or derivatives, such as artemisinin and metformin[1,2]. Berberine (BBR) is the main active component of the ancient Chinese herb *Coptis chinensis*. For thousands of years, this herb has been used in traditional Chinese medicine to treat diabetes. The glucose-lowering effect of BBR has been validated in both animal experiments[3] and human clinical trials[4]. However, the detailed mechanisms regarding the antidiabetic effect of BBR, especially whether it promotes insulin secretion, remain poorly understood, limiting the clinical use of BBR.

Glucose-stimulated insulin secretion (GSIS) from pancreatic islet β-cells is regulated by the electrogenic pathway associated with ATP-sensitive $K^+$ ($K_{ATP}$) channels. High glucose metabolism leads to an increase in the intracellular ATP/ADP ratio and subsequently causes closure of $K_{ATP}$ channels, depolarization of the plasma membrane, calcium influx through voltage-dependent calcium channels (VDCCs), and exocytosis of insulin secretory granules. $K_{ATP}$ channels have been used as targets of insulin secretagogues, including sulfonylureas and glinides. Over the past half century, these drugs have been used as the main oral antidiabetic drugs to promote insulin secretion in patients. These agents directly block $K_{ATP}$ channels and stimulate insulin secretion in a high-glucose-independent manner, which can cause life-threatening hypoglycemia[5].

Discovery of disease-associated genetic mutations in human can lead to discovery of novel biology, which may be therapeutic targets, and ion channels may well be one such pathway, as in the case of sulfonylureas and glinides[6,7]. Recently, we identified the pivotal role of KCNH6, a voltage-dependent $K^+$ ($K_v$) channel, in insulin secretion, by studying a large four-generation family with monogenic diabetes[8]. GSIS from pancreatic islet β-cells is influenced by high-glucose-dependent repolarization caused by $K_v$ channels such as KCNH6[9].

In this work, we find that blockade of KCNH6 channels by BBR increases insulin secretion in a high-glucose-dependent manner or only under hyperglycemic conditions, suggesting that BBR is a glucose-dependent insulin secretagogue for the treatment of diabetes that does not cause hypoglycemia.

## Results

**BBR increases high-glucose-dependent insulin secretion.** Inadequate insulin secretion in response to glucose is the main and early pathogenesis of diabetes. BBR is a plant alkaloid isolated from the *Coptidis* rhizoma, the dried rhizome of *Coptis chinensis* (Fig. 1a). The indicated concentrations of BBR were nontoxic to cell viability (Fig. 1b). We detected the insulinotropic effects of several doses of BBR on primary cultures of mouse pancreatic islets to examine the role of BBR in insulin secretion (Fig. 1c). At a low (2.8 mM) glucose concentration, BBR did not affect insulin secretion. In contrast, in the presence of a high (25 mM) glucose concentration, insulin release was dose-dependently stimulated by BBR. These data indicate that the effect of BBR on insulin secretion depends on a high glucose concentration. This glucose-dependent insulin secretion was further confirmed by an islet perifusion system with dynamic changes in glucose levels ranging from low to high. Mouse pancreatic islets perifused with BBR exhibited higher insulin secretion than those perifused with vehicle (control) in both the first phase (1–5 min) and second phase (6–30 min) (Fig. 1d). Because islet β-cells exposed to low to high glucose concentrations require at least 1 h to return to the resting state[10], BBR was applied after 2 h of high-glucose perfusion to ensure that the islets had returned to the steady state.

Stimulation of the islets with 30 mM KCl after treatment with BBR served as the internal control for islet viability. Insulin secretion was remarkably increased when BBR was applied during high-glucose perifusion (Fig. 1e).

We assessed insulin secretion by applying glibenclamide (a $K_{ATP}$ inhibitor) and verapamil (a VDCC blocker) combined with BBR to further validate the effect of BBR on glucose-stimulated insulin secretion and determine whether this effect depended on the $K_{ATP}$ channel and/or VDCC. As shown in Fig. 1f, g, BBR further promoted insulin secretion from isolated islets after glibenclamide treatment, while the effect of BBR was completely abolished by verapamil. Thus, the target of BBR was independent of $K_{ATP}$ and dependent on intact VDCC activity.

**BBR increases insulin secretion in wild-type but not *Kcnh6* knockout mice.** Previous studies have suggested a more significant glucose-lowering effect of BBR on diabetic animal models, e.g., *db/db* hyperglycemic mice or mice with high-fat diet (HFD)-induced hyperglycemia, than in normoglycemic mice[11,12]. Recently, we reported the key role of the KCNH6 potassium channel in insulin secretion and generated *Kcnh6* knockout (KO) diabetic mice[8]. Considerable changes were not detected in body weight, body temperature, respiratory exchange rate (RER), and energy expenditure (Supplementary Fig. 1a-d) between HFD-fed wild-type (WT) and KO mice. HFD-fed *Kcnh6* KO mice showed overtly impaired glucose tolerance (Supplementary Fig. 1e) and decreased insulin secretion (Supplementary Fig. 1f), without any change in insulin sensitivity compared with WT mice (Supplementary Fig. 1g).

BBR was administered to both HFD-fed hyperglycemic WT mice and HFD-fed hyperglycemic *Kcnh6* KO mice to better examine the effect of BBR on treating diabetes. BBR significantly decreased blood glucose levels (Fig. 2a) and increased serum insulin levels (Fig. 2b) in HFD-fed hyperglycemic mice after glucose loading, as evidenced by improved results for the intraperitoneal glucose tolerance test (IPGTT). Surprngly, in HFD-fed hyperglycemic *Kcnh6* KO mice (Fig. 2c), BBR did not significantly improve glucose homeostasis (Fig. 2d) or insulin secretion (Fig. 2e). Tolbutamide, a sulfonylurea insulin secretagogue, was tested to determine whether the islet insulin secretion function was still present in hyperglycemic *Kcnh6* KO mice. After tolbutamide treatment, the KO mice exhibited significantly improved glucose homeostasis (Supplementary Fig. 2a) and a corresponding increase in insulin secretion (Supplementary Fig. 2b). Based on the results, hyperglycemic *Kcnh6* KO mice secrete more insulin upon stimulation with insulin secretagogues other than BBR.

**BBR does not promote insulin secretion in the β-cell-specific KO mice.** Pancreatic islet β-cell-specific *Kcnh6* knockout (βKO) mice were generated using the Cre-LoxP recombinase system to further verify the KCNH6-dependent effects of BBR on insulin secretion (Fig. 2h and Supplementary Fig. 3). *Kcnh6* βKO mice were both viable and fertile and showed no abnormalities in body weight (Supplementary Fig. 4a). *KCHN6* gene is in the same *KCNH* family of another gene, *KCNH2 (HERG)*, and *HERG* is known to induce the Long QT syndrome[13]. We, thus, examined the cardiac phenotype of both global KO mice and βKO mice. No difference was observed in electrocardiogram (ECG) results compared to the control mice (Supplementary Fig. 5). While the global KO mice exhibited hyperglycemia in later adulthood[8], the βKO mice exhibited higher glucose levels (Supplementary Fig. 4b) and lower plasma insulin concentrations (Supplementary Fig. 4c) after glucose loading in early adulthood (8 weeks) after the consumption of a normal chow diet. The detailed mechanism

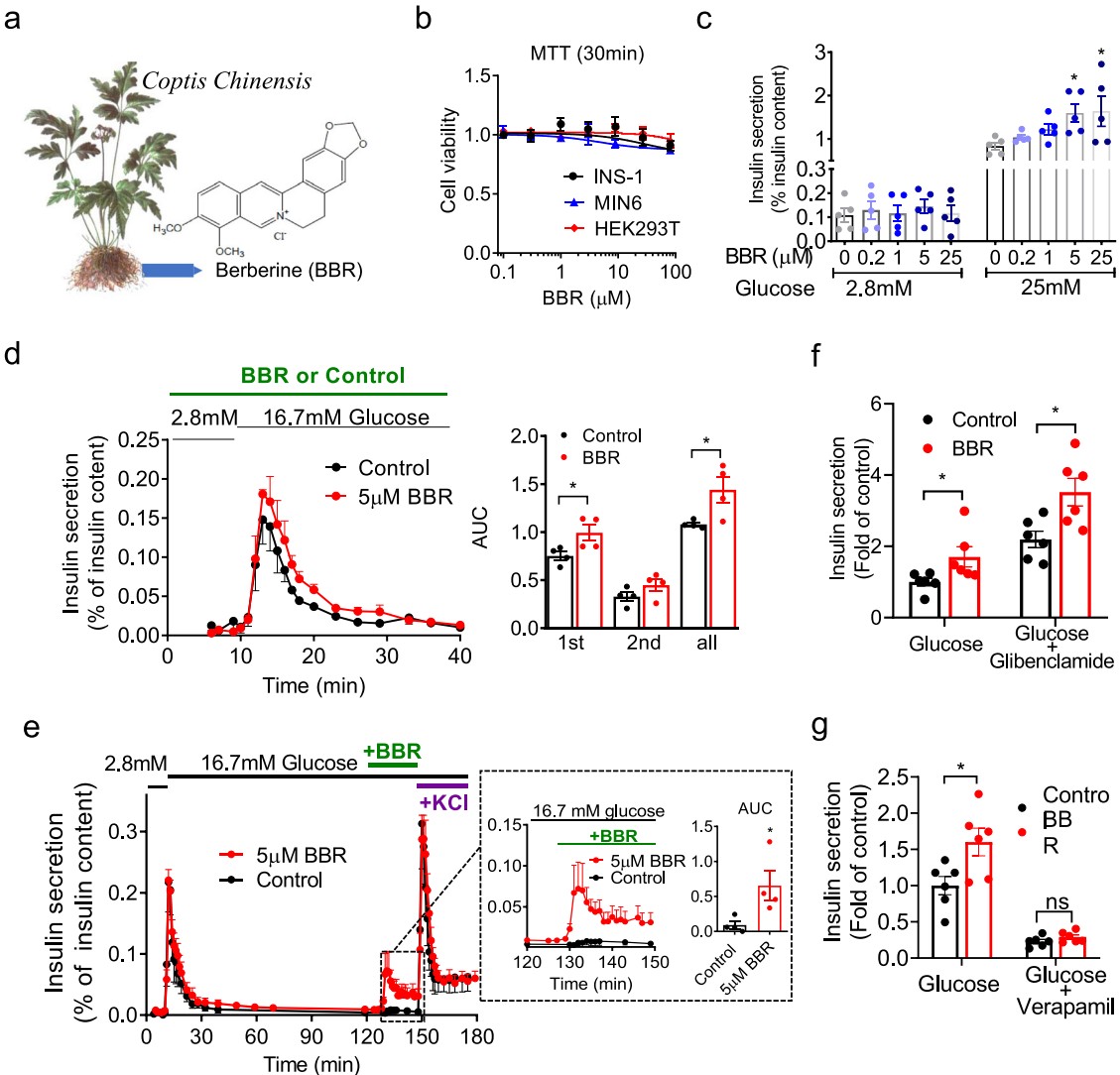

**Fig. 1 BBR increases high-glucose-dependent insulin secretion from pancreatic islets. a** Berberine (BBR) is a plant alkaloid isolated from the *Coptidis* rhizoma, the dried rhizome of *Coptis chinensis*. **b** Viability of the indicated cells incubated with different concentrations of BBR for 30 min. $n = 5$. **c** Mouse islets (15 islets per batch) were incubated with low (2.8 mM) or high (25 mM) glucose concentrations in the absence or presence of the indicated concentrations of BBR. The secreted insulin level in the supernatant was normalized to the corresponding insulin content of the islets. $n = 5$. $P = 0.0417$ for 5 μM BBR, $P = 0.0297$ for 25 μM BBR under 25 mM glucose concentration. Statistical significance was assessed using one-way ANOVAs coupled with Dunnett's post-hoc test (two-sided). **d** In the absence (control) or presence of 5 μM BBR, ~50 islets from male C57BL/6 J mice were perfused with a low (2.8 mM) glucose concentration for 10 min followed by a high (16.7 mM) glucose concentration for 30 min. The amount of insulin secreted was normalized to the total insulin content. First phase (0–5 min), second phase and total insulin secretion were calculated as the area under the curve (AUC). $n = 4$. $P = 0.042$ for first phase, $P = 0.036$ for total insulin secretion. Statistical significance was assessed using the Mann–Whitney $U$-test (two-sided). **e** Approximately 50 islets were perfused with a low (2.8 mM) glucose concentration for 10 min and then with a high (16.7 mM) glucose concentration for 120 min. Afterwards, during the continuous perfusion of 16.7 mM glucose, the islets were treated with 5 μM BBR or vehicle (control) for 20 min followed by 30 mM KCl, which served as the internal control for islet viability, for 30 min. The amount of insulin secreted was normalized to the total insulin content. The dotted box represents insulin secretion from islets stimulated with 5 μM BBR or vehicle under high (16.7 mM)-glucose conditions. $n = 4$. $P = 0.029$. Statistical significance was assessed using the Mann–Whitney $U$-test (two-sided). **f, g** Mice islets were treated with or without 5 μM BBR or BBR plus 1 μM glibenclamide (**f**) /100 μM verapamil (**g**) in the presence of 25 mM glucose for 30 min. Insulin secretion was normalized to that of the control group. $n = 6$. $P = 0.042$ for glucose, $P = 0.015$ for glucose plus glibenclamide in **f**. $P = 0.025$ for glucose, $P = 0.171$ for glucose plus verapamil in **g** Statistical significance was assessed using the Mann–Whitney $U$-test (two-sided). The values are presented as means ± s.e.m. *$P < 0.05$.

underlying the phenotypic discrepancies was unclear but recent studies on gene compensation[14,15] may provide an insight to explain this finding. *Kcnh6* βKO mice exhibited normal insulin sensitivity (Supplementary Fig. 4d). We examined the effects of BBR on insulin section in control and βKO mice at early adult ages after the consumption of a normal chow diet to avoid biased effects on insulin secretion caused by serious β-cell failure in later adulthood. Similar to the results obtained from HFD-fed *Kcnh6*

KO mice (Fig. 2d, e), BBR significantly decreased blood glucose levels (Fig. 2f) and increased the plasma insulin concentration (Fig. 2g) after high-dose (3 g/kg bw) glucose loading in the normoglycemic control mice. In contrast, it had no effects on glucose homeostasis (Fig. 3d) or insulin release (Fig. 3e) in the βKO mice.

We evaluated GSIS from the islets of control and βKO mice in vitro to still further confirm the β-cell-specific effects of BBR on KCHN6 channels and insulin secretion. Compared to the

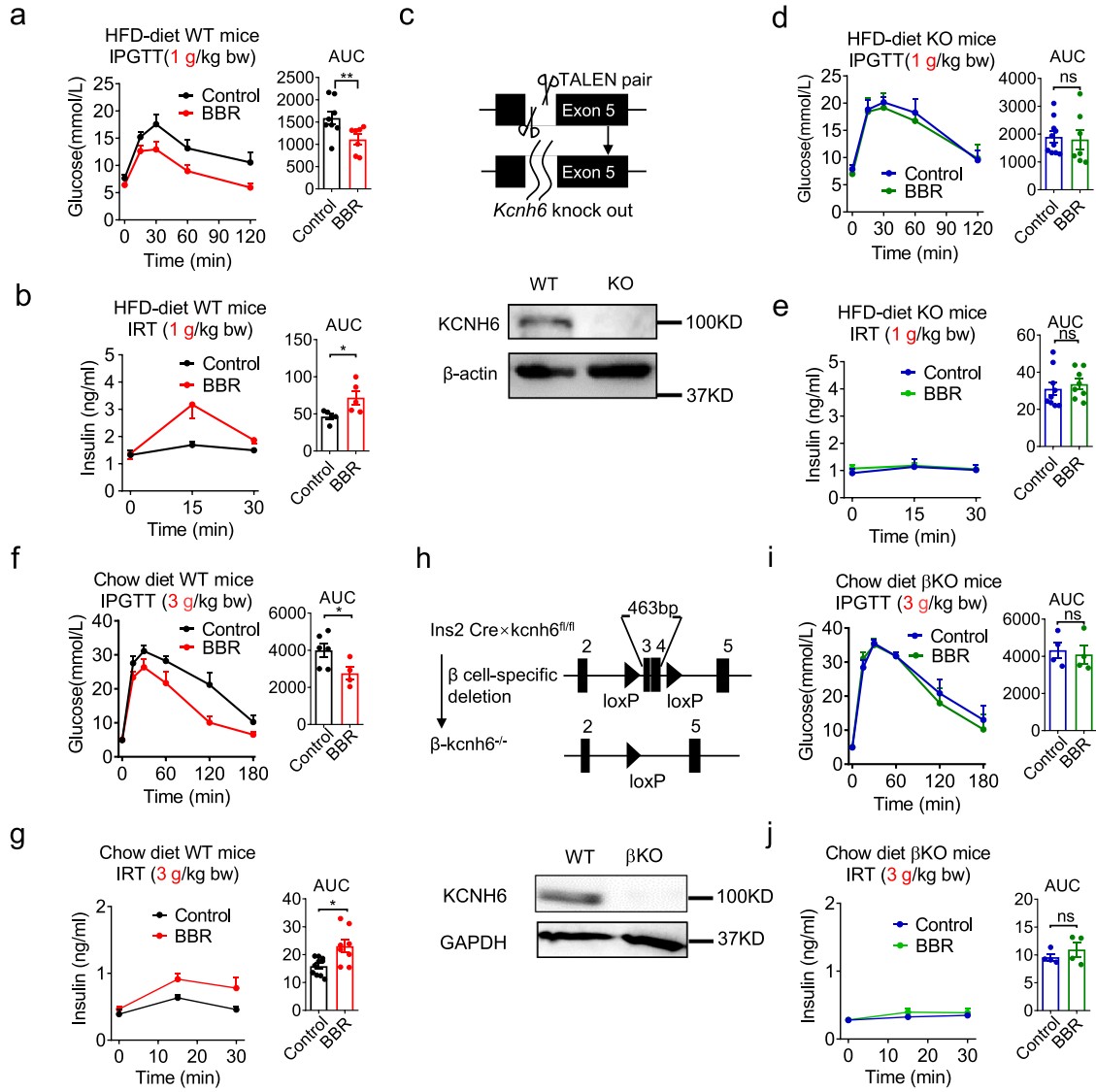

**Fig. 2 BBR increases insulin secretion in high-fat diet (HFD)-fed hyperglycemic mice but not in *Kcnh6* global knockout (KO) or *Kcnh6* islet β-cell-specific knockout (βKO) hyperglycemic mice. a–e** HFD-fed mice and *Kcnh6* KO mice were orally administered 560 mg/kg BBR or vehicle before being loaded with 1 g/kg glucose for the intraperitoneal glucose tolerance test (IPGTT) and insulin release test (IRT). **a** Blood glucose (control, $n = 8$; BBR, $n = 7$, $P = 0.009$) and **b** plasma insulin (control, $n = 5$; BBR, $n = 5$, $P = 0.016$) levels in HFD-fed mice. **c** *Kcnh6* KO mice were generated using the TALEN technique. Two TALENs specifically binding the target sequences in exon 5 of the *Kcnh6* gene resulted in the knockout of the gene. Western blot analysis of the KCNH6 protein from WT and KO mouse islets. A representative immunoblot from 3 different experiments is shown. **d** Blood glucose (control, $n = 10$; BBR, $n = 7$) and **e** plasma insulin (control, $n = 9$; BBR, $n = 8$) levels in *Kcnh6* KO mice. **f–j** Chow diet-fed control and *Kcnh6* βKO mice were orally administered 560 mg/kg BBR or vehicle before being loaded with 3 g/kg glucose for the IPGTT and IRT. **f** Blood glucose (control, $n = 6$; BBR, $n = 4$, $P = 0.038$) and **g** plasma insulin (control, $n = 10$; BBR, $n = 8$, $P = 0.012$) levels in chow diet-fed control mice. **h** *Kcnh6* βKO mice were generated using the CRISPR-mediated Cre-LoxP recombinase system. The endogenous *Kcnh6* locus was targeted for the conditional excision of exons 3 and 4. Western blot analysis of the KCNH6 protein from control and βKO mouse islets. A representative immunoblot from three different experiments is shown. **i** Blood glucose (control, $n = 4$; BBR, $n = 4$) and **j** plasma insulin (control, $n = 4$; BBR, $n = 4$) levels in *Kcnh6* βKO mice. The values are presented as means ± s.e.m. *$P < 0.05$ and **$P < 0.01$. Statistical significance was assessed using the Mann–Whitney *U*-test (two-sided).

vehicle control, BBR potentiated GSIS from islets obtained from control mice. However, the effect of BBR was not detected on islets from the βKO mice (Supplementary Fig. 6a).

The glucose-stimulated increase in the intracellular $Ca^{2+}$ concentration ($[Ca^{2+}]_i$), which directly triggers insulin secretion from β-cells, was also measured in primary cultures of pancreatic islet β-cells from control and βKO mice. While BBR significantly increased the $[Ca^{2+}]_i$ in pancreatic islet β-cells from control mice under hyperglycemic conditions, BBR did not significantly improve the response of the $[Ca^{2+}]_i$ to glucose in βKO pancreatic islet β-cells (Supplementary Fig. 6b).

*Kcnh6* knockdown (KD) INS-1 β-cells were generated with a specific shRNA (Supplementary Fig. 7a), and GSIS was evaluated in the KD and the control WT INS-1 cells. Although BBR significantly increased GSIS in the WT INS-1 cells, no differences were noted between BBR- and vehicle-treated KD INS-1 cells (Supplementary Fig. 7b).

Given the clear lack of effect of BBR on insulin secretion from pancreatic islet β-cells in the *Kcnh6* KO and βKO mice, in addition to *Kcnh6* KD INS-1 β-cells, KCNH6 channels on pancreatic islet β-cells must be involved in the effect of BBR to increase high-glucose-dependent insulin secretion.

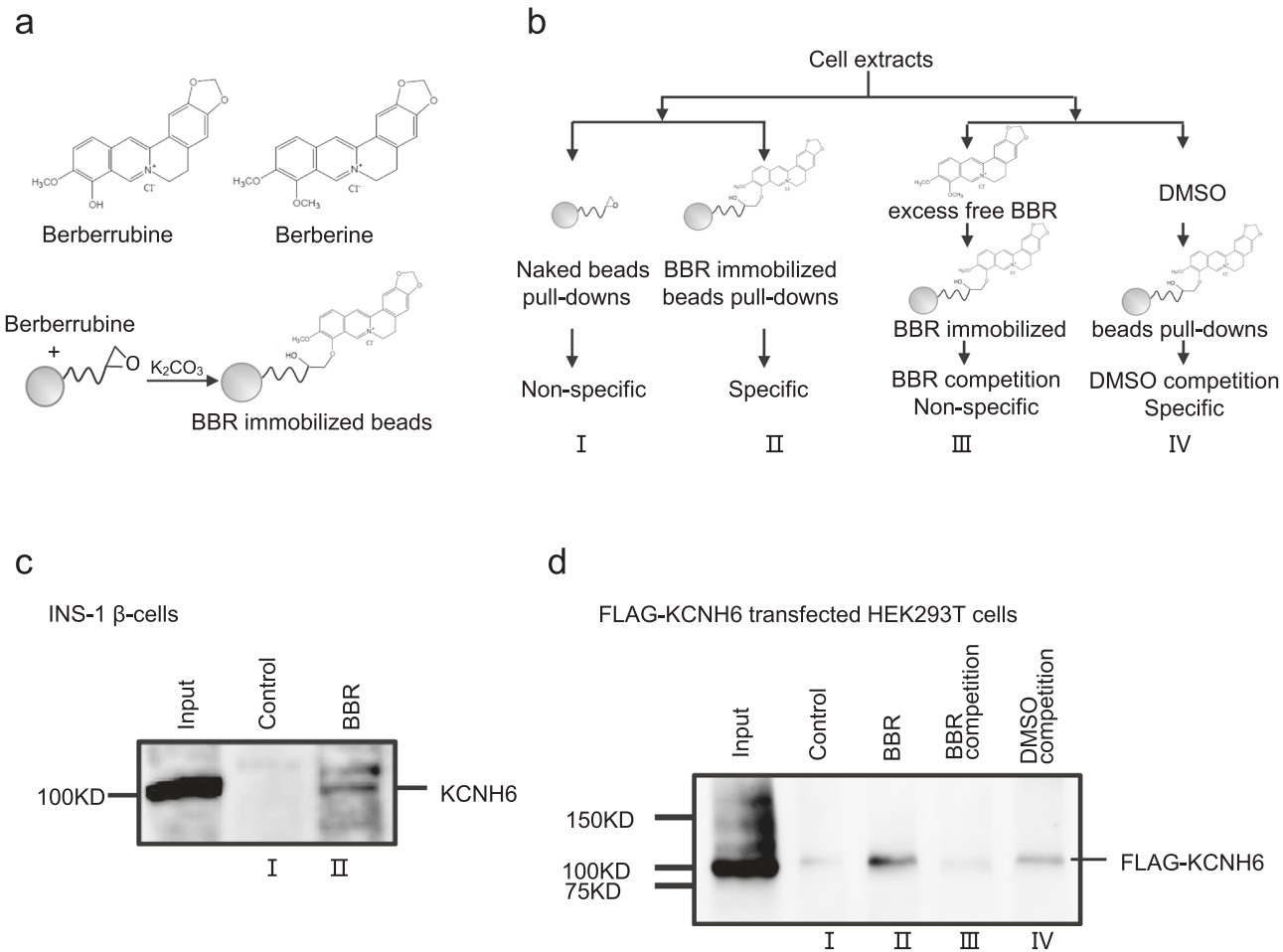

**Fig. 3 BBR targets KCNH6 channel proteins on islet β-cells. a** The active derivative of BBR, berberrubine, was immobilized on high-performance affinity beads. **b** Schematic of the experimental design. BBR-binding proteins were purified from cell extracts using different beads, as indicated. (I) Naked beads were used as a control; (II) BBR-immobilized beads were used to pull down BBR-binding proteins in the eluate; (III) 1 mM BBR was added to the extracts before the incubation with BBR-immobilized beads to reduce the yield of specific binding proteins; (IV) the same amount of vehicle (DMSO) was added to cell extracts before the incubation with the beads as a control for (III). Bound proteins were eluted and subjected to western blot analyses. **c** BBR-binding proteins from INS-1 β-cell extracts were analyzed with western blotting using an anti-KCNH6 antibody. A representative immunoblot from three different experiments is shown. **d** KCNH6-null human embryonic kidney 293 T (HEK293T) cells were transfected with plasmids encoding FLAG-tagged human *KCNH6*. BBR-binding proteins from transfected HEK293T cell extracts were analyzed with western blotting using an anti-FLAG antibody. A representative immunoblot from three different experiments is shown.

**BBR binds to the KCNH6 channel protein on pancreatic islet β-cells.** We used high-performance affinity beads that allowed the purification of drug-targeted proteins[16] from cell extracts to determine whether BBR targeted the KCNH6 protein on pancreatic islet β-cells. The active BBR derivative berberrubine was covalently conjugated to the beads (Fig. 3a). Pull-down experiments were performed using naked or BBR-immobilized beads (Fig. 3b I-II). Fractions eluted from the beads were subjected to immunoblotting and probed with a KCNH6-specific antibody. KCNH6 was clearly isolated as a BBR-specific binding protein in INS-1 β-cells (Fig. 3c).

KCNH6-null human embryonic kidney 293 T (HEK293T) cells were transfected with plasmids encoding FLAG-tagged human *KCNH6* to confirm the identity of the eluted protein. Cell extracts were isolated, and the pull-down experiments were repeated in the presence or absence of competing BBR in the solution (Fig. 3b III-IV). In the presence of competing BBR, the potential target protein bound to the BBR in the solution, and the amount of target protein binding to the beads would be reduced. When competing BBR was added to the reaction system, the yields of the KCNH6 were significantly reduced compared to those

observed in the absence of BBR (Fig. 3d). These results convincingly show that BBR specifically and directly interacts with the KCNH6 protein molecules.

**BBR inhibits KCNH6 currents by accelerating channel closure.** Patch-clamp recordings were performed to examine whether BBR binding to the KCNH6 channel affects channel function. We transfected *KCNH6*-null HEK293T cells (a model cell line commonly used to express specific ion channels expressed at low levels or non-endogenous ion channels, including $K_v$ channels, on the cell membrane) with the human *KCNH6* gene (Fig. 4a).

The current density (Fig. 4b-d) of KCNH6 was evaluated to measure overall KCNH6 function. KCNH6 channel currents were evoked by a specific voltage-step stimulus, showing KCNH6 current characteristics (Fig. 4b). Representative recordings of current traces from untransfected, *KCNH6*-transfected and *KCNH6*-transfected plus BBR-treated HEK293T cells are shown (Fig. 4b). Exposure to BBR resulted in a significant decrease in the step (Fig. 4c) and tail (Fig. 4d) current densities, suggesting that BBR reduced the KCNH6 current density in *KCNH6*-transfected HEK293T cells. The dose-response relationship showed that BBR

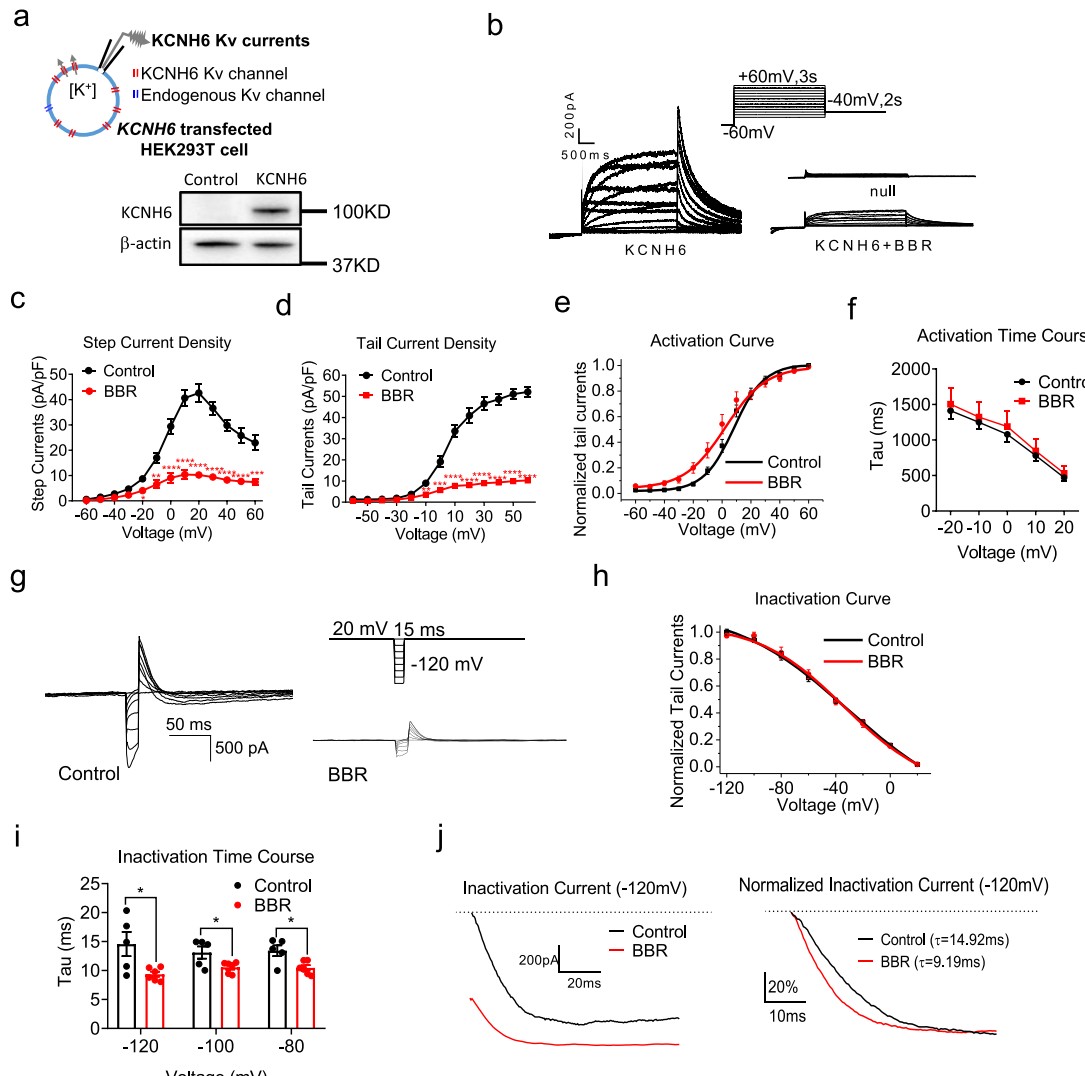

**Fig. 4 BBR inhibits KCNH6 currents by accelerating channel inactivation. a** KCNH6 currents were recorded from transfected HEK293T cells with majority of KCNH6 $K_v$ channels and very small amount of endogenous $K_v$ channels using a specific stimulus protocol. A representative immunoblot from three different experiments is shown. **b–f** Activation of the KCNH6 channel. **b** KCNH6 $K_v$ currents in the untransfected cells and *KCNH6*-null HEK293T cells transfected with the human *KCNH6* gene treated with 10 μM BBR. **c** I–V curve for step currents measured in the steady state. $P = 0.013$ for −20 mV, $P = 0.001$ for −10 mV, $P < 0.0001$ from 0 to 60 mV. **d** I–V curve for tail currents, as measured from the peak currents. $P = 0.005$ for −10 mV, $P = 0.0002$ for 0 mV, $P < 0.0001$ from 10 to 60 mV. **e** Comparison of the steady-state activation curves. The lines represent the best fit to the data according to a Boltzmann curve. **f** Time constants of activation ($n = 8$). **g–j** Inactivation of the KCNH6 channel. **g** Representative inactivation currents. **h** Comparison of steady-state inactivation curves. The lines represent the best fit to the data according to a Boltzmann curve ($n = 8$). **i** Time constants of inactivation ($n = 8$). $P = 0.026$ for −120 mV, $P = 0.037$ for −100 mV, $P = 0.016$ for −80 mV. **j** Representative traces of inactivation tail currents triggered by a −120 mV stimulus. The right panel shows the normalized inactivation traces. The time constants of inactivation are 14.92 ms and 9.19 ms for control and BBR-treated cells, respectively. The values are presented as means ± s.e.m. *$P < 0.05$, **$P < 0.01$, ***$P < 0.001$, and ****$P < 0.0001$. Statistical significance was assessed using the Mann–Whitney U-test (two-sided).

reduced KCNH6 currents with an $IC_{50}$ of $1.01 \pm 0.04$ μM (Supplementary Fig. 8). These results further confirm that BBR decreases overall KCNH6 function, as indicated by the KCNH6 current density.

The current kinetics (Fig. 4e–j) of ion channel gating are often altered by its blockers. Therefore, we examined the steady-state activation curves (Fig. 4e) and activation time courses (Fig. 4f) of the KCNH6 channels treated with BBR or vehicle (control). The activation curve of KCNH6 exhibited a slight leftward shift (Fig. 4e), indicating that BBR slightly altered the voltage-dependent channel activation. The activation time increased slightly after BBR treatment, indicating BBR slightly delayed channel opening (Fig. 4f). Then, the inactivation currents of

KCNH6 were recorded to evaluate the channel inactivation kinetics (Fig. 4g). The inactivation curve remained unchanged after BBR treatment (Fig. 4h). Noticeably, BBR significantly shortened the inactivation time, suggesting that BBR remarkably accelerated channel closure (Fig. 4i–j). Taken together, BBR slightly alters the activation kinetics while significantly accelerating KCNH6 channel inactivation (closure).

The overall KCNH6 channel function is mainly evaluated by measuring two parameters: the electrophysiological kinetics of channels and the number of channels on the plasma membrane[17]. Therefore, the number of endogenous KCNH6 channels on the plasma membrane was measured by isolating the membrane fraction of INS-1 β-cells. Compared to vehicle-treated cells, BBR

treatment neither affect the expression of the KCNH6 protein on the plasma membrane nor the synthesis of the protein (Supplementary Fig. 9). Based on these results, BBR does not alter the number of KCNH6 channels on the plasma membrane.

Thus, BBR directly binds the KCNH6 channel, significantly accelerates channel closure, and subsequently inhibits KCNH6 channel currents.

**BBR inhibits KCNH6 currents and promotes insulin exocytosis.** Primary islet β-cells were isolated from WT and KO mice to examine if the inhibitory effects of BBR on the KCNH6 channel affect the electrophysiological activities of pancreatic islet β-cells. As KCNH6 is one of the $K_v$ class channels expressed in islet β-cells, the whole-cell total $K_v$ currents were recorded (Fig. 5a). $K_v$ currents were elicited by depolarizing pulses in voltage-clamp mode (Fig. 5b). The current density, which represents overall channel function, was calculated (Fig. 5c-e). Consistent with our previous findings[8], the proportion of $K_v$ currents in the pancreatic islet β-cells from *Kcnh6* KO mice was significantly reduced compared to that in islet β-cells from WT mice (Fig. 5c). When BBR was administered to the β-cells of WT mice, $K_v$ currents were reversibly suppressed upon washout of BBR (Fig. 5d). Notably, BBR did not exert a significant effect on $K_v$ currents in the β-cells from the *Kcnh6* KO mice (Fig. 5e), suggesting that BBR suppressed $K_v$ currents (as indicated here as the currents through KCNH6 channels) or KCNH6 currents. In addition, these differential $K_v$ current responses to BBR were observed between WT and *Kcnh6* KD INS-1 β-cells (Supplementary Fig. 10a–d). These results clearly reveal the direct inhibitory effect of BBR mainly on KCNH6 $K_v$ currents in pancreatic islet β-cells.

$K_v$ channels are mainly responsible for the repolarization of the action potential in excitable cells, including pancreatic islet β-cells. Blockade or loss of function of $K_v$ channels may prolong the action potential duration (APD) due to impaired repolarization[18]. Electrical action potentials were evoked in current-clamp mode to examine whether BBR treatment prolongs the APD in pancreatic islet β-cells[19] (Fig. 5f). Repolarization of the action potential to the resting level took longer in KO β-cells than in WT β-cells (Fig. 5g) and in the presence of BBR than in its absence (Fig. 5h). The APD was recovered in WT pancreatic islet β-cells after the washout of BBR (Fig. 5h). However, no change in APD was observed in BBR-treated KO pancreatic islet β-cells or after washout of BBR (Fig. 5i). The resting potential and action potential amplitude and depolarization time were not changed by BBR among all different groups (Supplementary Fig. 11). The APDs of the WT and *Kcnh6* gene-silenced INS-1 β-cells (Supplementary Fig. 10e-f) were altered in a similar manner to those of the WT and KO pancreatic islet β-cells. Taken together, these findings suggest that the prolongation of the APD in pancreatic islet β-cells by BBR is mainly mediated by the inhibition of KCNH6 currents.

We examined the dynamic motion of single insulin granules by performing a total internal reflection fluorescent (TIRF) microscopic analysis to explore the direct effect of the BBR treatment on insulin exocytosis in pancreatic β-cells (Fig. 5j–l). The total number of insulin granules that were exocytosed after BBR treatment was considerably increased compared with vehicle-treated cells (Fig. 5k–l). Thus, BBR directly and acutely promotes insulin exocytosis in β cells under high-glucose conditions.

**BBR increases insulin secretion in humans.** We performed a pilot phase 1 clinical trial (Clinical Trials ID: NCT03972215) in healthy volunteers using the hyperglycemic clamp technique to assess the effect of BBR on insulin secretion in humans. This technique is the gold standard for assessing the response of pancreatic islet β-cells to high glucose concentrations[20,21]. Fifteen healthy male subjects with a normal glucose tolerance (Supplementary Table 1) were studied in a randomized, double-blind, placebo-controlled, two-period crossover clinical trial. Previous literatures studying the PK parameters of BBR showed that after single oral dose of BBR, the $t_{1/2a}$ was 0.869–0.87 h, $t_{max}$ was 2.37–4.0 h[22–24]. Previous clinical studies have evaluated BBR at dosages ranging from 600 to 2700 mg per day for as long as 6 months in adults[25–28]. The most common and most effective dosage appears to be 500 mg three times a day. As studies that evaluating the effects of insulin secretagogues usually used a single dose of the drug[29–32], a single dose of 1 g of BBR tablets or placebo tablets was administered orally. Then, a 160 min hyperglycemic clamp was performed 1 h after BBR administration at a baseline blood glucose level of +6.9 mmol/L as the target level of hyperglycemia[33]. All subjects participated in experiment on 2 days separated by a 14-day washout period. Blood samples were collected during the hyperglycemic clamp study to determine the plasma insulin and proinsulin C-peptide concentrations following treatment with BBR or the placebo (Fig. 6a).

Steady-state plasma glucose levels during the hyperglycemic clamp study were comparable between the BBR and placebo groups at each timepoint (Fig. 6b and Supplementary Table 2). Plasma insulin (Fig. 6c) and proinsulin C-peptide (Fig. 6g) responses to the hyperglycemic clamp were biphasic, peaking within 5 min, decreasing, and then increasing until the end of the 160 min period[29,33]. Total (Fig. 6d) and second phase (Fig. 6f) insulin secretion increased ~40% in the subjects treated with BBR. Similarly, total (Fig. 6h) and second phase (Fig. 6j) C-peptide levels were significantly higher in the BBR group than in the placebo group. First phase insulin (Fig. 6e) and C-peptide (Fig. 6i) secretion were also higher in the BBR group than in the placebo group, but the differences did not reach statistical significance; this difference might be caused by the previously observed high variability in incremental first phase values[29]. Notably, compared with the placebo, BBR did not change fasting blood glucose (FBG), fasting insulin or fasting proinsulin C-peptide levels (Supplementary Table 2). All subjects tolerated BBR well, and we observed no side effects. We also monitored ECG after the study and we did not observed difference between placebo and BBR groups (Supplementary Fig. 12). These findings led us to conclude that BBR significantly increases insulin secretion under hyperglycemic conditions but does not affect basal glucose levels in humans.

## Discussion

In this study, we provided evidence that unlike currently used insulin secretagogues that directly promote insulin secretion regardless of the blood glucose levels, such as sulfonylureas and glinides, BBR effectively lowered blood glucose levels only under hyperglycemic conditions. BBR bound to the KCNH6 protein and accelerated KCNH6 channel closure. It subsequently inhibited outward $K_v$ currents through KCNH6 channels and delayed the repolarization of pancreatic islet β-cells after high-glucose-stimulated action potentials, resulting in prolonged APDs. Prolonged APDs resulted in an increase in $Ca^{2+}$ influx during glucose stimulation. Our in vitro and in vivo studies showed that these BBR-induced electrogenic changes were immediately mirrored by increased insulin secretion (Fig. 7).

According to previous studies, inhibition of $K_v$ currents prolongs APD and promotes GSIS from pancreatic islet β-cells[9,34], suggesting that $K_v$ channels may represent new and better anti-diabetic drug targets. Inhibition of $K_v$ channels enhances the increase in $[Ca^{2+}]_i$ after glucose-mediated $K_{ATP}$ closure and

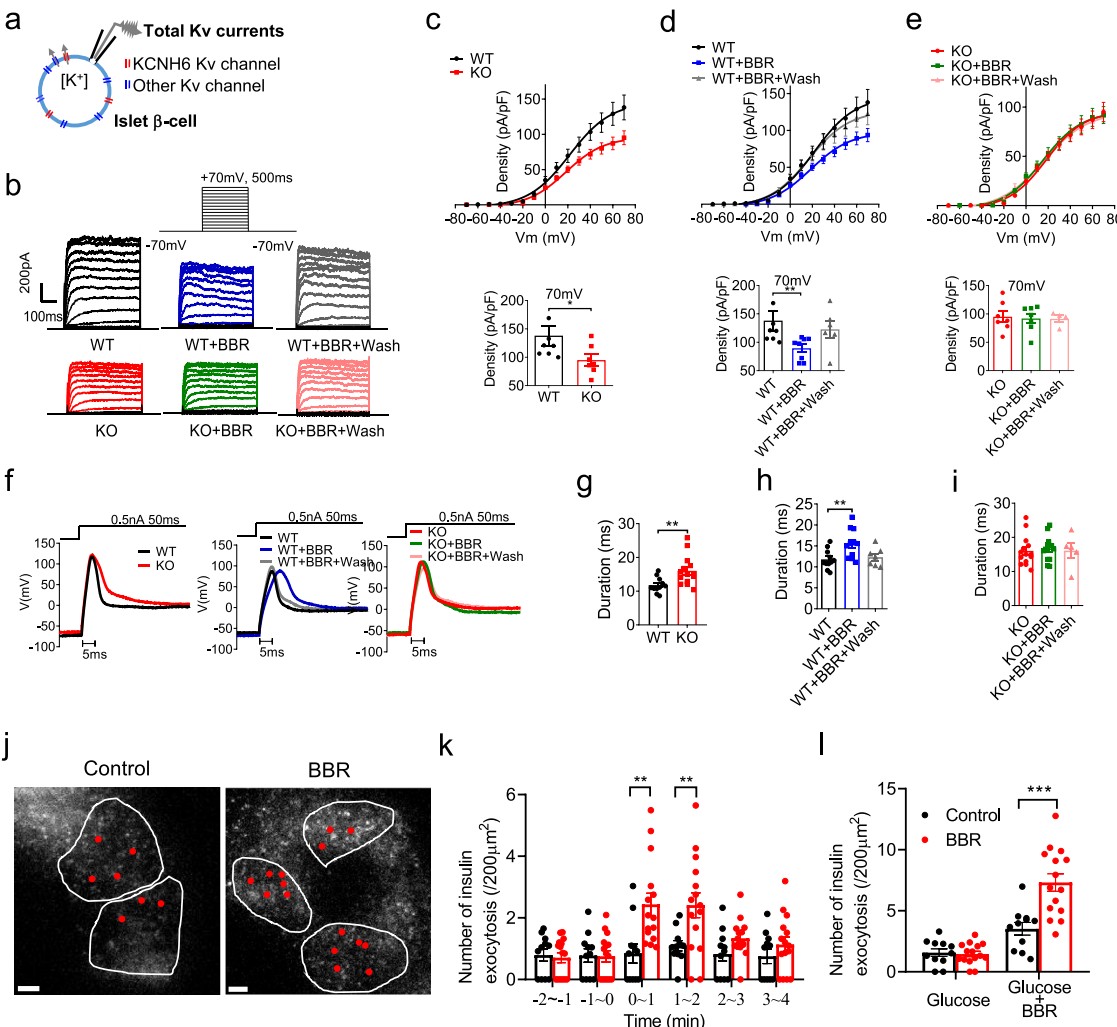

**Fig. 5 BBR inhibits voltage-dependent potassium (K$_v$) currents, prolongs action potential durations (APDs), and promotes insulin exocytosis in mouse pancreatic islet β-cells. a–e** Total K$_v$ currents in WT and *Kcnh6* KO mouse pancreatic islet β-cells treated with 10 μM BBR or vehicle plus washout. **a** The total K$_v$ currents of pancreatic islet β-cells expressing endogenous KCNH6 K$_v$ channels and other endogenous K$_v$ channels were measured in voltage-clamp mode. **b** Representative K$_v$ currents were recorded from the indicated WT and KO β-cells. **c–e** Summary of the steady-state current-voltage (I–V) curves for K$_v$ currents (upper panels) and the mean K$_v$ current densities at +70 mV (lower panels) (WT, $n = 8$; WT + BBR, $n = 8$; WT + BBR + wash, $n = 6$; KO, $n = 7$; KO + BBR, $n = 7$; KO + BBR + wash, $n = 4$). $P = 0.029$ in (**c**), $P = 0.023$ in (**d**). **f–i** Action potentials in WT and KO pancreatic islet β-cells treated with 10 μM BBR or vehicle plus washout. **f** Representative action potentials were recorded from the indicated WT and KO β-cells in current-clamp mode. **g–i** Summary of the mean APDs (WT, $n = 11$; WT + BBR, $n = 11$; WT + BBR + wash, $n = 7$; KO, $n = 14$; KO + BBR, $n = 13$; KO + BBR + wash, $n = 5$). $P = 0.005$ in **g**, $P = 0.008$ in **h**. **j–l** Total internal reflection fluorescence (TIRF) microscopy analysis of pancreatic β-cells from WT mice. Islet β-cells expressing insulin-EGFP were incubated with 25 mM glucose for 30 min followed by the application of 10 μM BBR or control (DMSO). **j** TIRF images show live β-cells treated with the control or BBR. Red dots indicate the positions of exocytotic events occurring over 4 min after the application of BBR or the control. The white lines represent the outline of cells. Scale bars, 5 μm. A representative image from three different experiments is shown. **k, l** The average numbers of fusion events in control ($n = 11$ cells) and BBR-treated cells ($n = 15$ cells) at 1 min intervals are shown in **k** and are summed in **l**. $P = 0.004$ for 0–1 min, $P = 0.004$ for 1–2 min in **k**, $P = 0.0005$ in **l**. The values are presented as means ± s.e.m. *$P < 0.05$, **$P < 0.01$, and ***$P < 0.001$. Statistical significance was assessed using the Mann–Whitney *U*-test (two-sided).

action potentials. Therefore, K$_v$ blockers promote high-glucose-dependent insulin secretion without affecting fasting insulin or glucose levels. This finding is consistent with the observed effects of BBR on glucose-dependent insulin secretion. In fact, stimulating high-glucose-dependent insulin secretion is a much better strategy for treating diabetes, as it is associated with a significantly lower risk of hypoglycemia, a very serious clinical problem in diabetes treatment[35]. Unfortunately, to date, no research on the use of K$_v$ inhibitors for treating patients with diabetes has been reported.

In this study, we reported that BBR, which targeted the KCNH6 K$_v$ channels, served as the first K$_v$ inhibitor to treat humans with diabetes. A 40% increase in plasma insulin levels under modest hyperglycemia was achieved in 15 healthy subjects who received a single oral dose of BBR in the hyperglycemic clamp study. Compared to sulfonylureas and DPP-4 inhibitors, which have been shown to maximally increase insulin levels by 120% and 50%[29,36], respectively, the effects of BBR appear moderate. This study is a pioneering trial in which a one-time single dose of BBR was administered and a modest target glucose level (blood glucose levels were increased 6.9 mmol/L from baseline) was used for the hyperglycemia clamp procedure. Larger insulinotropic effects of BBR may be observed after multiple treatments or by increasing target glucose levels for the

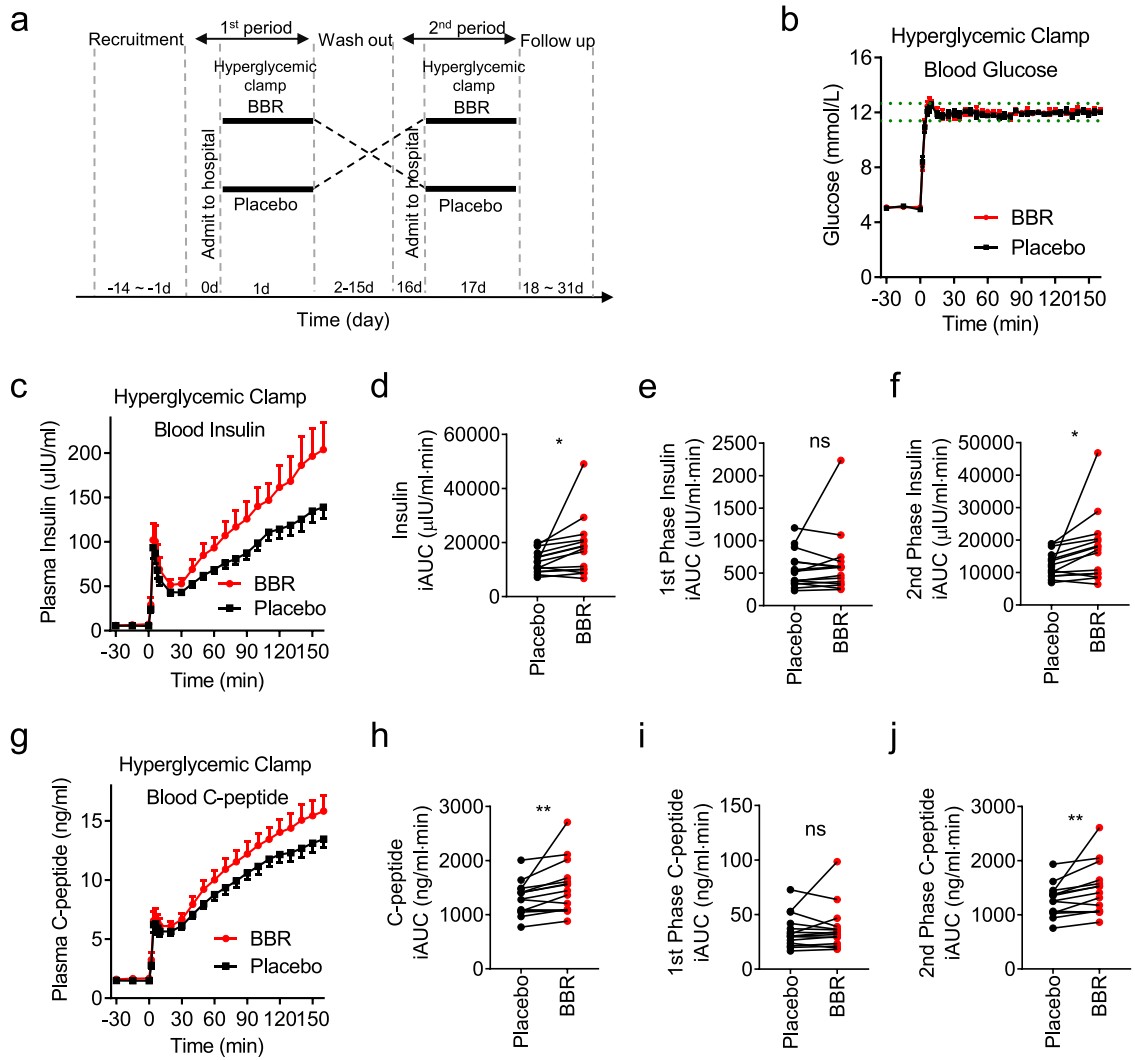

**Fig. 6 BBR increases insulin secretion in humans. a** A hyperglycemic clamp study was performed in 15 healthy male subjects using a crossover study design. Volunteers were examined and recruited according to the criteria within 14 days before the experiment (−14 d to −1 d). All subjects were admitted to the hospital on 0 d and underwent a 160-min hyperglycemic clamp after a single oral administration of 1 g of BBR or placebo on the next morning (1 d). All subjects participated in experiments on two days separated by a 14-day washout period (2–15 d) and changed the drug (BBR or placebo) in the next experiment. After the last experiment, subjects were followed up for 14 days to observe the side effects (18–31 d). **b** Plasma glucose levels in subjects throughout the clamp study. The dashed lines represent the range of ±5% of the hyperglycemic (basal blood glucose level +6.9 mmol/L) target level. **c** Insulin levels throughout the clamp study. $n = 15$. **d–f** Incremental plasma insulin AUC (insulin iAUC) throughout **d** the whole period (0–160 min), **e** the first phase (0–10 min), and **f** the second phase (10–160 min). **g** Proinsulin C-peptide levels throughout the clamp study. $n = 15$. $P = 0.024$ in **d** and $P = 0.022$ in **f**. **h–j** Incremental plasma proinsulin C-peptide AUC (C-peptide iAUC) throughout **h** the whole period (0–160 min), **i** the first phase (0–10 min), and **j** the second phase (10–160 min). $n = 15$. $P = 0.007$ in **h** and $P = 0.006$ in **j**. The values are presented as means ± s.e.m. *$P < 0.05$, and **$P < 0.01$. Statistical significance was assessed using the ratio paired $t$-test (two-sided).

hyperglycemic clamp procedure. This hypothesis was supported by our findings that BBR did not alter insulin secretion under low- or normal-glucose conditions in humans or mice. We postulate that the insulinotropic effects of BBR may be more robust in the presence of severe hyperglycemia.

In China, BBR has been extensively used as a nonprescription drug to manage diarrhea since the 1950s. The large-scale use of BBR confirms its good safety in clinical practice. Additionally, in the present study, BBR was well tolerated by all subjects in the clinical trial.

As a natural compound, BBR may exert effects on multiple targets in different cell types[37–40]. This study mainly focused on pancreatic β-cells and suggested that BBR specifically targeted KCNH6 channel in β-cells to promote insulin secretion. Although most papers reported positive effect of BBR on

insulin secretion[41–51], some papers reported negative effects on insulin secretion after the administration of BBR[52–55]. The opposite results might be due to different experimental conditions, especially a lower purity of BBR in vitro. As BBR is a natural product purified from plants, the components of this product are extraordinarily complex and may include some harmful impurities.

BBR has been reported to improve insulin resistance through the AMPK and the PKA pathways[11,56,57]. In fact, even insulin secretagogues, including sulfonylureas and glinides[58,59], ameliorate or relieve insulin resistance in patients with diabetes. Although metformin and glucagon-like peptide-1 (GLP-1) analogues, as well as insulin secretagogues, also affect the AMPK and PKA pathways[60–64], none of these compounds target the AMPK/PKA pathway directly. Therefore, the anti-insulin-resistance

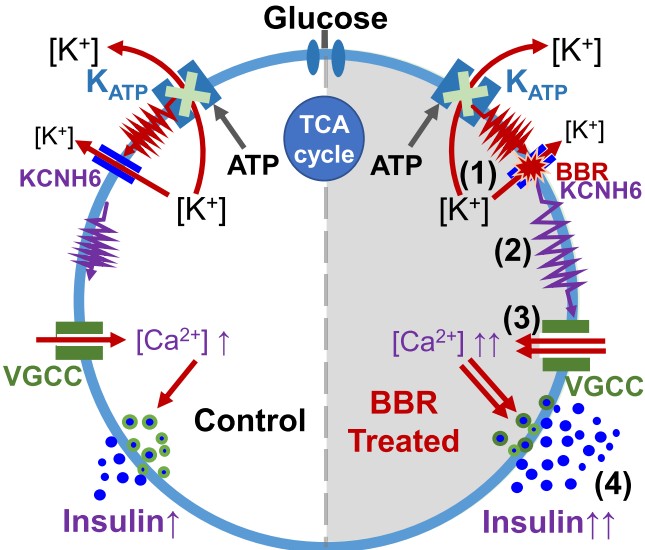

**Pancreatic Islet β-cell Insulin secretion**

**Fig. 7 Proposed mechanisms by which BBR increases insulin secretion.**
(1) By directly binding to the KCNH6 channel and accelerating channel closure, BBR blocks outward $K_v$ channel currents and prolongs APDs in pancreatic islet β-cells. (2) The prolongation of APDs results in an increase in $Ca^{2+}$ influx through voltage-gated calcium channels (VGCCs), (3) resulting in the accumulation of intracellular $Ca^{2+}$. (4) Increased intracellular $Ca^{2+}$ concentrations trigger more insulin secretion. Therefore, BBR may have potential as a new drug treatment for diabetes.

effects of BBR may be partially attributed to the glucose-lowing effects of this drug.

The long-term effect of BBR on insulin secretion via blocking the KCNH6 channel is a reasonable concern, partly because a loss-of-function mutation of the *KCNH6* gene causes both hyper-insulin secretion at a young age and hypo-insulin secretion in adulthood[8]. This finding may be explained by the complete loss-of-function of KCNH6 channel that induced continuous hyper-insulin secretion with cellular ER stress, apoptosis and subsequent loss of pancreatic islet β-cell mass and hypo-insulin secretion[8]. However, the KCNH6 mutation is different from BBR-induced KCNH6 blockade. Mutation will lead to lifelong dysfunction of the KCNH6 channel, while the effect of BBR on blocking KCNH6 channel is temporary. Actually, $K_{ATP}$ channel gene mutations also cause hyper-insulinemic hypoglycemia in infancy and diabetes in early adulthood[6,7,65,66]. $K_{ATP}$ channel blockers such as sulfonylureas, insulin secretagogues, have been widely used to treat diabetes for over 60 years without pro-diabetic effects. Similar to $K_{ATP}$ channel blockers such as sulfonylureas, BBR, a blocker of the KCNH6 channel, may be used to treat diabetes without pro-diabetic effects. Long-term use of BBR would relieve β-cells from the stress of hyperglycemia. Furthermore, since BBR only functioned under high-glucose conditions, it would be better than $K_{ATP}$ channel blockers because it would not cause hypoglycemia.

In conclusion, this study provides the first evidence that BBR prolongs high-glucose-induced APDs in pancreatic islet β-cells by inhibiting KCNH6 $K_v$ channels. BBR increases high-glucose-dependent insulin secretion in humans and is an excellent insulin secretagogue without the potential side effect of causing hypoglycemia. In addition, this study provides information on KCNH6 channels as a target for the development of better and safer antidiabetic drugs.

## Methods

**Materials.** Tolbutamide, glibenclamide, and verapamil were purchased from Sigma–Aldrich (St. Louis, MO, USA). Berberine (PHR1502, Lot#LRAA9232) used in the animal and in vitro studies was purchased from Sigma–Aldrich. For the hyperglycemic clamp study, tablets of BBR (Lot#23180501) and the placebo (Lot#180401) used in human subjects were admitted by the State Food and Drug Administration (SFDA) of China and were obtained from Xin Yi Tian Ping Pharmaceutical Co., Ltd. (Shanghai, China). Antibodies used for immunoblotting and immunofluorescence staining are listed in Supplementary Table 3.

**Animal model and animal care.** C57BL/6 J mice were purchased from Vital River Laboratories (Beijing, China). The mice were housed at constant temperature and humidity, with a 12 h light and dark cycle, and fed a regular unrestricted diet. As indicated, a HFD (60% fat, 20% carbohydrate, and 20% protein in calorie percentage; Research Diets, New Brunswick, NJ, USA) was provided to mice beginning at 4 weeks of age for 8 weeks. Both male and female mice were used in our experiment, and we did not observed any sex-based differences. The *Kcnh6* knockout (KO) mice were generated as described previously[8]. The homozygous offspring were identified by polymerase chain reaction (PCR) and DNA sequencing. The *Kcnh6* β-cell-specific knockout (βKO) mice were generated as described below. The endogenous *Kcnh6* (PubMed Gene ID: 192775) locus was targeted for the conditional excision of exons 3 and exon 4. A targeting vector was designed to contain upstream/downstream homology arms and a LoxP-flanked central region (exons 3-4). LoxP (ATAACTTCGTATAATGTATGCTATACGAAGTTAT) sites were specifically inserted into introns upstream of exon 3 and downstream of exon 4 using standard gene targeting protocols. Two small guide ribonucleic acids (sgRNAs) targeting the introns on both sides of the floxed region of *Kcnh6* and the targeting vector were, respectively, constructed and transcribed in vitro. Then, the Cas9 mRNA, sgRNAs, and targeting vector were co-injected into zygotes. Afterwards, the zygotes were transferred into the oviduct of pseudopregnant ICR female mice at 0.5 days post-coitus (dpc). F0 mice were born 19–21 days after transplantation, and all the offspring of ICR female mice (F0 mice) were identified by PCR and sequencing of tail DNA. Positive F0 mice were genotyped using the aforementioned methods. Finally, F0 mice were crossed with C57BL/6 J mice to increase the number of heterozygous mice. Transgenic mice expressing Cre recombinase under the control of a β-cell-specific rat insulin 2 promoter, Rip-Cre mice, were a gift from Professor Aimin Xu, Hongkong University. The mice were derived from a C57BL/6 J genetic background. β-Cell-specific inactivation of *Kcnh6* was achieved by crossbreeding Rip-Cre transgenic mice with mice carrying the floxed *Kcnh6* gene. In all experiments, βKO mice were sacrificed at the age of 6–8 weeks and littermates from the same breeding pair were used as controls.

Animal experiments followed the national ethical guidelines implemented by our Institutional Animal Care and Use Committee and were approved by the Ethical Review Committee of the Institute of Zoology, Capital Medical University, China.

**Cell culture and transfection.** All cells were cultured in a humidified incubator with 95% air and 5% $CO_2$ at 37 °C. INS-1 832/13 β-cells (Cell Resource Center, Chinese Academy of Medical Sciences, Beijing, China) were cultured in RPMI 1640 medium containing 10% FBS, 1 mM L-glutamine, 1 mM HEPES, 1 mM sodium pyruvate, 50 μM 2-mercaptoethanol, and 1% penicillin–streptomycin. HEK293T cells (Cell Resource Center, Chinese Academy of Medical Sciences, Beijing, China) were cultured in DMEM containing 10% FBS and 1% penicillin–streptomycin. Pancreatic islets and primary pancreatic islet β-cells were cultured in RPMI 1640 medium supplemented with 10% FBS and 1% penicillin–streptomycin. Pancreatic islets were isolated from mice euthanized by cervical dislocation through the injection of 500 units/mL collagenase solution (type XI; Sigma–Aldrich, St. Louis, MO, USA) into the pancreatic duct, followed by digestion at 37 °C for 25 min with mild shaking, and isolated islets were picked by hand selection under a dissecting microscope, as described previously[67,68]. After isolation and recovery, pancreatic islets were treated with 0.05% trypsin in PBS for 5 min at 37 °C and fully dispersed to isolate primary pancreatic islet β-cells. *Kcnh6* knockdown INS-1 β-cells were transfected with the *Kcnh6* shRNA or scrambled shRNA (as a negative control) using Lipofectamine 2000 (Invitrogen, South San Francisco, CA, USA). The following target sequence was used: 3'-ACACGCA-GATGCTGCGTGTCAAGGAGTTC-5'. The WT *KCNH6* (PubMed Gene ID: 81033) cDNA was subcloned into pcDNA3-FLAG and GV314-EGFP vector as previously described[8]. HEK293T cells were transiently transfected using Lipofectamine 2000 reagent (Invitrogen) according to the manufacturer's protocol.

**Cell viability assay.** Cells (INS-1, MIN6, and HEK293T) were incubated with 0.1, 0.3, 1, 3, 9, 27, or 81 μM BBR for 30 min. Effects of BBR on cell viability and cytotoxicity were determined using the 3-(4,5-dimethylthiazol-2-yl)-2,5-diphenyltetrazolium bromide (MTT) assay.

**GSIS and islet perfusion secretory assay.** INS-1 832/13 β-cells at passage numbers 6-11 were used for the insulin secretion assay. Cells were seeded onto 24-well plates at a density of $5 \times 10^5$ cells per well. Static incubation insulin release from intact islets was monitored using 10 islets from each batch. After 24 h of

culture, cells or islets were balanced with standard low-glucose Krebs-Ringer buffer (KRB) containing 120 mM NaCl, 5 mM KCl, 2 mM CaCl$_2$, 1 mM MgSO$_4$, 24 mM NaHCO$_3$, 0.1% BSA, 15 mM HEPES (pH 7.4), and 2.8 mM glucose for 60 min at 37 °C. After balancing, cells or islets were first incubated with 2.8 mM KRB buffer for 30 min. Then, these cells were incubated with KRB buffer containing 16.7 mM or 25 mM glucose and the indicated concentrations of BBR, indicated drugs or vehicle (DMSO) for another 30 min at 37 °C. Secreted insulin levels were measured using an AlphaLISA insulin kit with an EnVision 2101 multilabel reader (Perki-nElmer Life Sciences, San Jose, CA, USA). Results were normalized to the total insulin content.

The perfusion of isolated islets was executed. Briefly, ~50 islets were placed at the bottom of a 1 mL syringe that had been cut to a volume of 400 µl and plugged with cotton. Islets were perfused with standard low-glucose KRB at a constant flow rate of 1.0 mL/min for 30 min. After this stabilization period, they were further perfused with the same buffer for 10 min followed by different medium containing the indicated secretagogues. Fractions were collected every 1 min, and all islets were collected after the perfusion assay for an assessment of the insulin content. Insulin secretion was measured using an AlphaLISA insulin kit with an EnVision 2101 multilabel reader (PerkinElmer Life Sciences, San Jose, CA, USA). Results are presented as the amount of insulin secreted normalized to the islet insulin content.

**IPGTT, intraperitoneal insulin release test (IRT), and insulin tolerance test (ITT)**. WT and *Kcnh6* KO mice and control and *Kcnh6* βKO mice were assigned to weight- and sex-matched groups and received BBR (560 mg/kg body weight)[11] or vehicle by gavage 1 hr before the intravenous injection of glucose to reach the maximum blood concentration of BBR. In some experiments, tolbutamide (40 mg/kg body weight) or vehicle were administered by gavage before the intravenous injection of glucose. The IPGTT, IRT, and ITT were performed as described previously[8].

**Mice ECG analysis**. Both male and female mice (~20 week) were used in this experiment, littermates from the same breeding pair were used as controls. Mice were intraperitoneally anesthetized using pentobarbital. ECG analysis was performed in bipolar configuration using three needle-type electrodes (one positive, one negative, and one neutral or reference), positioned in the subcutaneous tissue. Electrodes were connected to a BioAmp amplifier (ADInstruments, New Zealand), analog signals were converted to digital signals through a Powerlab/8sp interphase system (ADInstruments, New Zealand), displayed, recorded, and analyzed on a personal computer using the LabChart 8.1.19 software (ADInstruments, New Zealand).

**Immunohistochemistry**. Pancreas tissues from WT and βKO mice were fixed with 4% paraformaldehyde (PFA). Tissues were transferred to 20% sucrose in PBS, incubated overnight at 4 °C, then transferred to 30% sucrose in PBS, and incubated overnight at 4 °C to impregnate fully. Samples were frozen and cut into 5–20 mm-thick sections and applied to slides. Sections were incubated with antibodies against insulin (1:200) and KCNH6 (1:100) in PBS containing 1% BSA in a humidified chamber overnight at 4 °C. Sections were washed with PBS and then incubated with the appropriate secondary antibodies diluted in PBS containing 1% BSA for 1 h at RT in the dark. Sections were washed with PBS, a coverslip was mounted with fluorescent mounting medium containing DPAI, and staining was examined using a fluorescence microscope immediately or sections were stored flat at 4 °C in the dark.

**Measurement of [Ca$^{2+}$]$_i$**. The [Ca$^{2+}$]$_i$ in primary pancreatic islet β-cells was determined using the Ca$^{2+}$-sensitive fluorescent indicator dye Fluo-4 (Sigma–Aldrich, St. Louis, MO, USA) according to the manufacturer's instructions. Baseline fluorescence was determined by averaging 20 images. For this experiment, 25 mM glucose with or without 10 µM BBR was added to the solution at 50 s during the observation. Images were recorded every 5 s.

**Preparation of BBR-immobilized beads**. The scheme of immobilization is shown in Fig. 3a. Magnetic FG beads (Linker beads, TAS8848N1110, Tamagawa Seiki, 1 mg) were incubated with a 10 mM solution of the berberine derivative berber-rubin in N,N-dimethylformamide and 14 mg of potassium carbonate for 16–20 h at 60 °C. Unreacted residues were masked using 50% methanol, and the resulting beads were stored at 4 °C.

**Affinity purification with BBR-immobilized beads**. BBR-immobilized beads (0.5 mg) were equilibrated with 100 mM KCl buffer containing 20 mM HEPES-NaOH (pH 7.9), 100 mM KCl, 1 mM MgCl$_2$, 0.2 mM CaCl$_2$, 0.2 mM EDTA,10% (v/v) glycerol, 0.1% NP-40, 1 mM DTT, and 0.2 mM PMSF. Cell extracts were prepared from transfected human 293 T cells and rat INS-1 β-cells as described and were incubated with the beads for 4 h at 4 °C. The beads were washed three times with 100 mM KCl buffer, and bound proteins were eluted with 1× loading dye solution containing 62.5 mM Tris-HCl (pH 6.8), 0.005% bromophenol blue, 2% SDS, 10% glycerol, and 5% 2-mercaptoethanol. In some experiments, 1 mM BBR was added to cell extracts before the incubation with the beads (Fig. 3b).

**Patch-Clamp experiments**. Whole-cell (conventional) patch-clamp experiments were performed as previously described[8]. Data were collected using PatchMaster v2x80 software. Recordings were performed before and at least 15 min after the application of 10 µM BBR or vehicle, as described elsewhere[69], and after fully washing out the compound in some experiments. K$_v$ currents were recorded from a holding potential of −70 mV to various depolarizing pulses (−70 to +70 mV) in 10 mV increments for 0.5 s in primary cultured pancreatic islet β-cells and for 2 s in INS-1 β-cells. Action potentials were elicited by a 0.5-nA current injection for 50 ms in primary cultured pancreatic islet β-cells and 0.1-nA current injection for 5 ms in INS-1 β-cells, as described elsewhere[70]. For recording KCNH6 channel currents, transfected HEK293 cells were elicited by 3-s depolarizing pulses ranging from −60 to +60 mV and the tail currents by 2-s repolarizing pulses to −40 mV. For activation curves, the tail currents were normalized to the peak values. Then, the normalized data were plotted against the pre-pulse potentials and fitted to the Boltzmann distribution: $I/I_{max} = 1/\{1+\exp[(V_{1/2}-V)/\kappa]\}$, where $I$ is the HERG tail current amplitude at a pre-pulse potential $V$, $V_{1/2}$ is the voltage for half-maximal activation, and $\kappa$ is a slope factor. A single exponential function was used to fit the curve of activation currents evoked at the step test potentials from a holding potential of −80 mV. The inactivation currents were elicited with a 2-s depolarizing pulse to +20 mV to inactivate the KCNH6 channels, followed by various repolarizing pulses to potentials from −120 mV to +20 mV for 15 ms, and then depolarized by a test pulse to +20 mV. In addition, the inactivation curves represented the best fits to the Boltzmann distribution: $I/I_{max} = 1/\{1+\exp[(V_{1/2}-V)/\kappa]\}$, where $I$ is the KCNH6 tail current amplitude at a pre-pulse potential $V$, $V_{1/2}$ is the voltage for half-maximal activation, and $\kappa$ is a slope factor. The decaying tail currents were fitted to a single exponential function to calculate the inactivation time course.

**Plasma membrane protein isolation and western blot**. INS-1 β-cells were treated with DMSO (vehicle) or the indicated concentration of BBR for 30 min. Plasma membrane (PM) proteins were isolated using the Minute$^{TM}$ Plasma Membrane Protein Isolation Kit (Invent Biotechnologies, Plymouth, MN, USA). Western blot experiments were performed as previously described[8].

**Total internal reflection fluorescence microscopy**. A monolayer of mouse β-cells was cultured on poly-L-lysine–coated 35-mm glass base dishes for 2 days. The cells were infected with an adenovirus encoding preproinsulin-EGFP (insulin-EGFP) and were further cultured for 48 hr. Total internal reflection fluorescence (TIRF) microscopy (the penetration depth of the evanescent field: 100 nm) was performed with a Nikon TIRF microscope system (Nikon, Tokyo, Japan) as described previously[71]. Sequential images were acquired at 101-ms intervals. Fusion events were manually counted and analyzed using a previously reported method[71].

**Phase 1 clinical trial using hyperglycemic clamp**. The study was in compliance with the CONSORT statement (Supplementary Information Note 1). The study was conducted with the approval of the Ethics Committee of Beijing Tongren Hospital, Capital Medical University (TRECKY2019-037). The study was compliant with the principles of the Declaration of Helsinki, current good clinical practice guidelines and all ethical regulations. The research plan was approved by the Ethics Committee of Beijing Tongren Hospital, Capital Medical University and is provided in the Supplementary Note 2. Written informed consent was obtained from each subject before participation in this study. The study setup was first registered at ClinicalTrials.gov (NCT03972215) on 05/29/2019. After minor editorial changes, final registration at ClinicalTrials.gov was obtained on the 05/31/2019. The first volunteer was included on the 10/01/2019, the last experiment was completed on the 01/20/2020 and the study was finished on the 02/17/2020 after the follow-up period and statistical analysis. The manuscript was written in compliance with ICMJE guidelines. All studies were performed in Beijing Tongren Hospital, Capital Medical University.

*Subjects*. Fifteen healthy male research subjects aged 18–45 yrs with a body mass index (BMI) of 18–25 kg/m$^2$, normal oral glucose tolerance, normal blood pressure, normal laboratory values for HbA1c and kidney and liver functions, no family history of diabetes, and no use of medications were enrolled in this study. The sample size was determined according to previous studies[29,36,72]. All volunteers were enrolled by the researchers. The baseline characteristics are presented in Supplementary Table 1.

*Study procedure and laboratory analysis*. This study employed a randomized, double-blind, placebo-controlled, two-period crossover design. The subjects were studied on two separate experimental days with an interval of 14 days apart, and randomized to receive BBR or the placebo. The random allocation was conducted by asking volunteers to select the randomized number generated by computer. Each number represented placebo or BBR, and each group has the same number of numbers. The subjects were instructed to maintain their usual lifestyle and did not drink alcohol or smoke cigarettes for 3 days before the study. On each experimental day, following an overnight fast, subjects received a single oral dose of 1 g of BBR or a matching dose of the placebo 1 h before the start of the clamp study. Fasting

blood glucose concentrations before and after drug treatment were measured to assess the effect of BBR on basal glucose levels. Fasting insulin and C-peptide levels were measured 1 h after the administration of BBR and before the glucose injection. Then, participants underwent a 160 min hyperglycemic clamp with a baseline blood glucose level +6.9 mmol/L as the target level. Two catheters were inserted into the antecubital veins of both arms for infusions and blood collection, and the arm was warmed to ~55 °C with a heating pad to obtain arterialized venous blood[73]. ECG was performed after the study. All subjects were followed up for 14 days to observe the side effects.

Blood samples used to determine glucose, insulin and proinsulin C-peptide levels were obtained at intervals throughout the clamp study. The blood samples for insulin and C-peptide determinations were obtained every 2 min within the first 14 min and every 10 min afterwards. Plasma glucose levels were measured every 5 min throughout the study (EKF Biosen C-Line Glucose and Lactate analyzer, Cardiff, UK). The hyperglycemic clamp was initiated with a 14 min priming dose of 20% glucose (Baxter, Shanghai, China) to quickly increase the blood glucose concentration to the target level, and then the glucose infusion rate (GIR) was adjusted to maintain the glucose level[33].

Glycated hemoglobin A1c (HbA1c) levels were measured using a high-performance liquid chromatography (HPLC) instrument, VARIANT II (Bio-Rad, California, USA). Insulin and C-peptide levels were determined using IMMUNLITE-2000 (Siemens, Malvern, PA, USA). These biochemical measurements were performed in the Chinese Ministry of Health Quality Assessment Program.

*Outcomes.* Primary outcomes were: (1) Differences of serum insulin levels between BBR and placebo treatment groups during the hyperglycemic clamp study. (2) Differences of serum C-peptide levels between BBR and placebo treatment groups during the hyperglycemic clamp study.

Secondary outcomes were: (1) Differences of glucose infusion rates between BBR and placebo treatment groups during the hyperglycemic clamp study. (2) Differences of blood glucose levels between BBR and placebo treatment groups during the hyperglycemic clamp study.

Other prespecified outcome was: Heart rate and QT-interval duration using electrocardiogram after drug treatment.

*Calculations.* First phase insulin and proinsulin C-peptide secretion were recorded as the mean incremental area under the curve (AUC) hormone responses during the first 10 min after the glucose injection. Second phase hormone secretion was recorded as the mean incremental insulin and proinsulin C-peptide responses from 10 to 160 min of the hyperglycemic clamp. The steady-state glucose level was calculated as the mean glucose concentration from 30 to 160 min in the study. The incremental AUC was defined as the AUC above the fasting level (mean of the values at time points −30, −15, and 0 min).

**Statistical analysis**. All statistical analyses were conducted with the software GraphPad Prism version 7.0 Data are presented as means ± s.e.m. Statistical significance was determined using unpaired Student's *t*-test or the Mann–Whitney *U*-test if n < 9, or one-way ANOVA as appropriate.

**Reporting summary**. Further information on research design is available in the Nature Research Reporting Summary linked to this article.

## Data availability

Data generated or analyzed during this study are included in this published article (and its supplementary information files and the Source Data file). Detailed volunteer-related study raw data, which could compromise protection of privacy of research participants are not publicly available due to privacy restrictions. These data are available in anonymized form from the corresponding author upon reasonable request. Source data are provided with this paper.

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

## Acknowledgements

We thank Ran Sun, Yanfang Du, Yuting Fu, and Lu Jin for their help in the hyperglycemic clamp study. We thank all the nurses from the Endocrinology department of Beijing Tongren Hospital for their help in the hyperglycemic clamp study. We thank all the volunteers in the hyperglycemic clamp study. We thank Jianping Feng for her assistance in lab management. This work was supported by grants from the National Key R&D Program of China (2017YFC0909600) and National Natural Science Foundation of China (81930019, 8151101058, and 81471014) to J.K.Y. The study was also supported by a grant from Kobayashi International Scholarship Foundation and MSD Scholarship Donation to T.I. This work was also supported by the Joint Research Program of the Institute for Molecular and Cellular Regulation, Gunma University (grants 17013 and 19003 to J.K.Y.). We thank Core Facilities Center of Capital Medical University for supporting this study. This study was registered at ClinicalTrials.gov under Registration No. NCT03972215.

## Author contributions

J.K.Y. conceived the idea for the study, designed the experiments, and wrote the manuscript. M.M.Z. designed and performed the experiments, and wrote the first draft of manuscript. S.L., J.L., H.W., X.C., Q.L, T.T.S., and K.M. performed some experiments, H.H.X., C.C., and T.I. helped with the interpretation of the results, design of experiments, and proofreading of the manuscript.

## Competing interests

The authors declare no competing interests.
