## [Peer Review File · Nature Communications]

Reviewers' Comments:

Reviewer #1:

Remarks to the Author:

This is an interesting and clearly presented manuscript describing the mechanistic basis of berberine action as an anti-diabetic agent. Although there is ample evidence linking berberine to insulin secretion, mechanistic links are unclear. The current manuscript seems to provide a clear link to berberine inhibition of KCNH6.

1. The results quite clearly point to an involvement of berberine via an effect on KCNH6 channels, since the effects are absent in KCNH6 KO mice. How the drug is acting seems much less clear though. The authors previously showed that KCNH6 KO leads to a diabetic phenotype in adult mice, although it enhanced insulin secretion in young animals. This is a complication worth speculating on - Since berberine inhibits the channels, this might seem to suggest that it should be pro-diabetic in older animals? The authors should also discuss the many papers that suggest alternate ways by which berberine affects insulin secretion and glucose metabolism. For example, there is evidence that glucose- and arginine-induced insulin secretion is increased by blockers of such channels (FASEB J . 2000 Dec;14(15):2601-10), and there is other evidence that berberine inhibits glucose oxidation and insulin secretion in rat islets (Endocr J . 2018 Apr 26;65(4):469-477). Can the authors comment on the decrease of secretion at high [berberine] concentration in Fig. 1b?

2. The binding experiments seem clear enough, but the localization experiments are less than convincing. It is far from clear that the cell image analyzed in Fig. 5 is meaningful, even within the same view there are cells that appear to show a very different distribution of intensity. Such experiments may be intrinsically flawed, since an overexpression system such as this is highly likely to cause build up of channels in physiologically relevant ER/Golgi compartments. Thus the interpretation that berberine alters surface expression is probably premature.

3. The action of berberine, if through the channels, is can apparently occur almost immediately, at least on a time scale of a few minutes (e.g. Fig. 1d). This does not really seem compatible with a trafficking effect. While the authors fail to show any kinetic effects on channel voltage dependence, this does not preclude some other mechanism that leads to chronic channel closure.

4. Berberine has been proposed to have effects on multiple other channels including inhibition of multiple potassium and calcium channels, through multiple mechanisms (e.g. J Pharmacol Sci. 2020 Apr;142(4):131-139; PLoS One. 2017 Aug 1;12(8):e0181823; Mol Med Rep. 2016 Oct;14(4):3985-91; Mol Med Rep. 2014 Sep;10(3):1576-82). The authors should discuss such studies in light of their claim of a very specific action here.

Reviewer #2:

Remarks to the Author:

Zhao et.al. in the manuscript entitled "Berberine, a Chinese herb-derived compound, as a novel insulin secretagogue targeting KCNH6 potassium channel" claimed that Berberine was a glucose dependent insulin secretagogue mediated by its effect on closing Kv channel encoded by Kcnh6. The complicated role of KCNH6 in regulating beta cell biology and hence glycemic regulations has been reported by the authors' previous study. It is new to know that KCNH6 might mediate the effect of BBR in beta cells which promotes insulin secretion. I think there are some serious problems in this study to qualify its publication in Nature communications.

First, there are several issues in the animal study which was lousy and ill-designed.

In the Figure 2, the insulin releasing test curve of HFD fed mice was uncommonly low and flat, according to our experience and ponderous published evidence. Thus, I am doubting if the HFD model is even successful, there are no data in this paper to verify this. And I am astonished that the author used global KO mice in HFD model and conditional KO mice in chow diet model, the metabolic phenotype compared to control in neither of the mouse models has been previously reported or described in the current study. The rationale under this kind of design is out of my knowledge.

Further, as to the Kcnh6KO mice, the authors did not offer the information of the animal age, according to the authors' own study that different age of Kcnh6KO mice have opposite metabolic phenotype. The dysfunction of KCNH6 causes overstimulation of insulin secretion in the short term and β cell failure in the long term. So, in which period the authors intended to test the effect of

BBR might affect the phenotype. Furthermore, the previous studies of the authors stated that the mechanism underlying the hyperglycemia Kcnh6KO mice is mainly relied on the increased beta cell ER stress and apoptosis, it would not be surprise that the dysfunctional beta cell could not respond to a weak secretagogue as BBR.

Second, how the BBR was dissolved, where it was purchased from or which kind of reagent they used as placebo in human subjects were not clarified in the manuscript. For, there are multiple evidence published about the effect of Berberine on regulating the insulin secretion, most of which are opposite to this study. Published reports have found the potential mechanisms relating to attenuated mitochondria oxidative phosphorylation via stimulating AMPK and inhibiting PKA signaling. The author should discuss and provide data how the BBR they used in this study affected the cell growth, which was the toxic dosage and discuss why the phenotype was opposite with most other studies.

Third, in Figure 5 b,c, the authors tried to show the changes of Kcnh6 localization in cell membrane, however the authors used the beta-actin as the loading control of the membrane portion, which is indicating the cell plasma not membrane portion.

Fourth, in Figure 6. details of the methodology for the clamp study were not explained clearly. For instance, Line 505-506, the authors just claimed that “--- insulin and proinsulin C-peptide were obtained at intervals throughout the clamp study”, but at what time interval, 5 minutes or longer? It is not mentioned in the Method or any other places in the manuscript. It seems to me that the time interval for the whole clamp study was all the same throughout the procedure. However, for most of the hyperglycemic clamp studies, the time intervals for insulin obtainment should be different between the first and second phase. Please clarify. And I also doubted the result of 6c-j. The differences between the calculated AUC value was much smaller than what have been presented in the insulin or C peptide curve between the treat and placebo group. I guess the significant differences might be resulted by an outlier presented in the scatter plots in Figure 6d-f or Figure 6h-j.

Most importantly, in all the experiments the authors conducted, BBR was used with a single dose, which is fine to observe in the cell electrophysiology changes or acute effect on insulin secretion, however, a single dosage before IPGTT, a single dosage before hyperglycemic clamp? I did not see the relevance of this acute effect of BBR in regulating beta cell biology as well as long term glucose homeostasis.

Last but not the least, I do not have question about the electrophysiology study the author conducted in BBR treated and Kcnh6KO cells. If the authors linked this alteration to the insulin secretion, I am wondering whether glucose induced membrane capacitance alteration measured by whole cell patch clamp was affected by BBR or loss of Kcnh6KO, which monitors bona fide insulin granule exocytosis.

It is not new that BBR exerts broad effects on metabolic organs such as liver, the adipose, endocrine pancreas and muscle tissue, even in gut microbiota. I am not convinced by the authors that the effect of BBR on Kv Channel in beta cell was linked to the phenotype of mice glucose excursion and beta cell secretion. Several gaps need to be filled and scientific significance and novelty of this study were also questionable.

Responses to the reviewers' comments

Reviewer #1 (Remarks to the Author):

This is an interesting and clearly presented manuscript describing the mechanistic basis of berberine action as an anti-diabetic agent. Although there is ample evidence linking berberine to insulin secretion, mechanistic links are unclear. The current manuscript seems to provide a clear link to berberine inhibition of KCNH6.

1. The results quite clearly point to an involvement of berberine via an effect on KCNH6 channels, since the effects are absent in KCNH6 KO mice. How the drug is acting seems much less clear though. The authors previously showed that KCNH6 KO leads to a diabetic phenotype in adult mice, although it enhanced insulin secretion in young animals. This is a complication worth speculating on - Since berberine inhibits the channels, this might seem to suggest that it should be pro-diabetic in older animals?

Answer: This question is very good and would be a cause for concern in most readers. As reported in our previous study ¹, KCNH6 dysfunction can cause hyper-insulin secretion at a young age and subsequent hypo-insulin secretion in early adulthood in both humans and mice. A potential explanation for this finding is that the non-functional KCNH6 channel increases insulin secretion in the early stage, along with ER stress and apoptosis, and subsequently decreases insulin secretion.

However, the *KCNH6* mutation is different from BBR-induced KCNH6 blockade. Mutation will lead to lifelong dysfunction of the KCNH6 channel, while the effect of BBR on blocking the KCNH6 channel is temporary. Actually, K_{ATP} channel gene mutations also cause hyperinsulinemic hypoglycemia in infancy and diabetes in early adulthood ²⁻⁵. K_{ATP} channel blockers such as sulfonylureas, insulin secretagogues, have been widely used to treat diabetes for over 60 years without *pro-diabetic* effects. Similar to K_{ATP} channel blockers such as sulfonylureas, BBR, a blocker of the KCNH6 channel may be used to treat diabetes without *pro-diabetic* effects.

We have added a discussion of this question in the paper (Page 15, Line 3 - 19).

1. Yang, J.K., et al. From Hyper- to Hypoinsulinemia and Diabetes: Effect of KCNH6 on Insulin Secretion. *Cell Rep* 25, 3800-3810 e3806 (2018).
2. Hugill, A., Shimomura, K., Ashcroft, F.M. & Cox, R.D. A mutation in KCNJ11 causing human hyperinsulinism (Y12X) results in a glucose-intolerant phenotype in the mouse. *Diabetologia* 53, 2352-2356 (2010).
3. Kapoor, R.R., et al. Hyperinsulinaemic hypoglycaemia and diabetes mellitus due to dominant ABCC8/KCNJ11 mutations. *Diabetologia* 54, 2575-2583 (2011).
4. Huopio, H., et al. A new subtype of autosomal dominant diabetes attributable to a mutation in the gene for sulfonylurea receptor 1. *Lancet* 361, 301-307 (2003).
5. Vieira, T.C., Bergamin, C.S., Gurgel, L.C. & Moises, R.S. Hyperinsulinemic hypoglycemia evolving to gestational diabetes and diabetes mellitus in a family carrying the inactivating ABCC8 E1506K mutation. *Pediatric diabetes* 11, 505-508 (2010).

The authors should also discuss the many papers that suggest alternate ways by which berberine affects insulin secretion and glucose metabolism. For example, there is evidence that glucose- and arginine-induced insulin secretion is increased by blockers of such channels (FASEB J. 2000 Dec;14(15):2601-10), and there is other evidence that berberine inhibits glucose oxidation and insulin secretion in rat islets (Endocr J. 2018 Apr 26;65(4):469-477).

Answer: Thank you for your suggestion. We searched PubMed for studies published in English up to February 14, 2021, with the search terms “((BBR) OR (Berberine)) AND (insulin secretion)”, we found **19** papers related to this topic ⁶⁻²⁴. Of these **19** papers, **11** studies showed a positive effect of BBR on promoting insulin secretion ^{7,10-12,15-21}. **Four** *in vivo* studies observed decreased insulin levels after long-term BBR administration due to decreased glucose concentrations ^{6,8,13,14}. Only **4** papers reported negative effects on insulin secretion after the administration of BBR ^{9,22-24}. The opposite results might be due to different experimental conditions, especially the lower purity of BBR *in vitro*. As BBR is a natural product purified from plants, the components of this products are extraordinarily complex, and may include some harmful impurities. We noticed that **3** (including the paper in *Endocr J. 2018*) of those **4** papers obtained BBR from National Institute for the Control of Pharmaceutical and Biological Products ^{9,23,24}. This institute does not produce any chemicals itself, and we are unable to find any information (constitution and purity) for BBR on the website of this institute (<https://www.nifdc.org.cn/>). Our BBR was purchased from Sigma-Aldrich (PHR1502, Lot#LRAA9232) with a high purity exceeding 99%. We confirmed increased insulin secretion from mouse islets *in vitro* and humans and mice *in vivo*.

We have added a discussion of this question (Page 14, Lines 11-19).

6. Cole, L.K., et al. Supplemental Berberine in a High-Fat Diet Reduces Adiposity and Cardiac Dysfunction in Offspring of Mouse Dams with Gestational Diabetes Mellitus. *The Journal of nutrition* (2021).
7. Li, J., et al. Amorphous solid dispersion of Berberine mitigates apoptosis via iPLA2beta/Cardiolipin/Opa1 pathway in db/db mice and in Palmitate-treated MIN6 beta-cells. *International journal of biological sciences* 15, 1533-1545 (2019).
8. Sun, Y., et al. Restoration of GLP-1 secretion by Berberine is associated with protection of colon enterocytes from mitochondrial overheating in diet-induced obese mice. *Nutrition & diabetes* 8, 53 (2018).
9. Bai, M., et al. Berberine inhibits glucose oxidation and insulin secretion in rat islets. *Endocrine journal* 65, 469-477 (2018).
10. Jiang, Y.Y., et al. Protective role of berberine and *Coptischinensis* extract on T2MD rats and associated islet Rin5f cells. *Molecular medicine reports* 16, 6981-6991 (2017).
11. Dong, Y., et al. Metabolomics Study of Type 2 Diabetes Mellitus and the AntiDiabetic Effect of Berberine in Zucker Diabetic Fatty Rats Using Uplc-ESI-Hdms. *Phytotherapy research : PTR* 30, 823-828 (2016).
12. Liu, L., et al. Uncoupling protein-2 mediates the protective action of berberine against oxidative stress in rat insulinoma INS-1E cells and in diabetic mouse islets. *British journal of pharmacology* 171, 3246-3254 (2014).
13. Perez-Rubio, K.G., Gonzalez-Ortiz, M., Martinez-Abundis, E., Robles-Cervantes, J.A. & Espinel-Bermudez, M.C. Effect of berberine administration on metabolic syndrome, insulin sensitivity, and insulin secretion. *Metabolic syndrome and related disorders* 11, 366-369 (2013).
14. Shan, C.Y., et al. Alteration of the intestinal barrier and GLP2 secretion in Berberine-treated type 2 diabetic rats. *The Journal of endocrinology* 218, 255-262 (2013).
15. Rayasam, G.V., et al. Identification of berberine as a novel agonist of fatty acid receptor GPR40. *Phytotherapy research : PTR* 24, 1260-1263 (2010).
16. Gao, N., Zhao, T.Y. & Li, X.J. The protective effect of berberine on beta-cell lipoapoptosis. *Journal of endocrinological investigation* 34, 124-130 (2011).
17. Yu, Y., et al. Modulation of glucagon-like peptide-1 release by berberine: in vivo and in vitro studies. *Biochemical pharmacology* 79, 1000-1006 (2010).
18. Lu, S.S., et al. Berberine promotes glucagon-like peptide-1 (7-36) amide secretion in streptozotocin-induced diabetic rats. *The Journal of endocrinology* 200, 159-165 (2009).
19. Wang, Z.Q., et al. Facilitating effects of berberine on rat pancreatic islets through modulating hepatic nuclear factor 4 alpha expression and glucokinase activity. *World journal of gastroenterology* 14, 6004-6011 (2008).
20. Ko, B.S., et al. Insulin sensitizing and insulinotropic action of berberine from *Cortidis rhizoma*. *Biological & pharmaceutical bulletin* 28, 1431-1437 (2005).
21. Leng, S.H., Lu, F.E. & Xu, L.J. Therapeutic effects of berberine in impaired glucose tolerance rats and its influence on insulin secretion. *Acta pharmacologica Sinica* 25, 496-502 (2004).
22. Lamontagne, J., et al. Pioglitazone acutely reduces insulin secretion and causes metabolic deceleration of the pancreatic beta-cell at submaximal glucose concentrations. *Endocrinology* 150, 3465-3474 (2009).
23. Zhou, L., et al. Berberine acutely inhibits insulin secretion from beta-cells through 3',5'-cyclic adenosine 5'-monophosphate signaling pathway. *Endocrinology* 149, 4510-4518 (2008).
24. Yin, J., et al. Effects of berberine on glucose metabolism in vitro. *Metabolism: clinical and experimental* 51, 1439-1443 (2002).

Can the authors comment on the decrease of secretion at high [berberine] concentration in Fig. 1b?

Answer: Thank you for your question. First, the indicated concentrations of BBR were non-toxic to β -cell viability (new Fig. 1b). Second, we re-analyzed the detailed procedure of the experiments and found that the vehicle (DMSO) concentrations were different in the experiments using different BBR concentrations. The higher BBR concentrations contained more DMSO. It may be the toxic effect of high DMSO but not high BBR concentration. We diluted the BBR stock solution into several concentrations and normalized the DMSO concentration to 0.05% in solutions with

the indicated BBR concentrations to avoid effects of DMSO. Under these conditions, we did not notice a decrease in insulin secretion at the highest BBR concentration (new Fig. 1c).

2. The binding experiments seem clear enough, but the localization experiments are less than convincing. It is far from clear that the cell image analyzed in Fig. 5 is meaningful, even within the same view there are cells that appear to show a very different distribution of intensity. Such experiments may be intrinsically flawed, since an overexpression system such as this is highly likely to cause build up of channels in physiologically relevant ER/Golgi compartments. Thus the interpretation that berberine alters surface expression is probably premature.

Answer: We appreciate your opinion. We initially examined the expression of endogenous KCNH6 on the plasma membrane 30 min after BBR administration, but failed to detect any difference (new Supplementary Fig. S8). Thus, short-term administration of BBR does not affect KCNH6 trafficking. Although the long-term BBR treatment of β -cells produced changes in the distribution of KCNH6 on the plasma membrane that might be due to the long-term effect of BBR on the trafficking or maturation of the KCNH6 protein, this long-term result was not suitable to interpret the effect of BBR on insulin secretion in this study. Since this paper mainly focused on the prompt effect of BBR on insulin secretion, we deleted the results for the long-term effect of BBR (original Fig. 5).

3. The action of berberine, if through the channels, is can apparently occur almost immediately, at least on a time scale of a few minutes (e.g. Fig. 1d). This does not really seem compatible with a trafficking effect. While the authors fail to show any kinetic effects on channel voltage dependence, this does not preclude some other mechanism that leads to chronic channel closure.

Answer: We agree with your suggestion, as indicated in our response to question 2. The action of BBR through KCNH6 occurred promptly. This effect does not appear compatible with a trafficking effect. In this version, we analyzed KCNH6 channel

kinetics and found that the BBR treatment shortened the time of channel inactivation (new Fig. 4i-j), suggesting that BBR significantly accelerated channel closure. Our data indicated that BBR directly bound to the channel, significantly accelerated channel closure, and subsequently inhibited KCNH6 channel K⁺ currents.

In this version, the electrophysiological kinetics were described as follows: Therefore, we examined the steady-state activation curves (Fig. 4e) and activation time courses (new Fig. 4f) of the KCNH6 channels treated with BBR or vehicle (control). The activation curve of KCNH6 exhibited a slight leftward shift (Fig. 4e), indicating that BBR slightly altered the voltage-dependent channel activation. The activation time increased slightly after BBR treatment, indicating BBR slightly delayed channel opening. Then, the inactivation currents of KCNH6 were recorded to evaluate the channel inactivation kinetics (Fig. 4g). The inactivation curve remained unchanged after BBR treatment (Fig. 4h). Noticeably, BBR significantly shortened the inactivation time, suggesting that BBR remarkably accelerated channel closure (new Fig. 4i-j). Taken together, BBR slightly alters the activation kinetics while significantly accelerating KCNH6 channel inactivation (closure) (Page 9, Lines 5 - 16).

4. Berberine has been proposed to have effects on multiple other channels including inhibition of multiple potassium and calcium channels, through multiple mechanisms (e.g. J Pharmacol Sci. 2020 Apr;142(4):131-139; PLoS One. 2017 Aug 1;12(8):e0181823; Mol Med Rep. 2016 Oct;14(4):3985-91; Mol Med Rep. 2014 Sep;10(3):1576-82). The authors should discuss such studies in light of their claim of a very specific action here.

Answer: Thank you very much for your questions. We agree that as a natural compound, BBR may exert effects on multiple other ion channels in different cell types, including ventricular myocytes, interstitial cells of Cajal and sinoatrial node cells²⁵⁻²⁸. Even a chemical compound with one target may act on different organs and result in different functions. For example, sulfonyleureas stimulate insulin secretion by blocking the K_{ATP}/SUR1 channel in islet β-cells, while it can also act on the

K_{ATP}/SUR2 channel in cardiac cells. This study mainly focused on the effect of BBR on insulin secretion from pancreatic β -cells.

We have added new data (new Fig. 1f-g) in the present version to support our conclusion of a specific effect of BBR on insulin secretion. As K_{ATP} and VDCC are two of the most important ion channels involved in insulin secretion, we used glibenclamide (a K_{ATP} inhibitor) and verapamil (a VDCC blocker) combined with BBR to identify the target of BBR in insulin secretion. As shown in new Fig. 1f-g, BBR further promoted insulin secretion after glibenclamide treatment, while the effect of BBR was completely abolished by verapamil. Based on these results, the target of BBR was independent of K_{ATP} and dependent on intact VDCC activity. Furthermore, BBR had no effect on insulin secretion under low-glucose conditions (Fig.1c), which also indicated that BBR did not affect VDCC and K_{ATP} channels to promote insulin secretion. In this study, KCNH6 contributed to ~40% of the total K_v currents (Fig. 5c) and a reasonable conclusion was that KCNH6 played a dominant (at least important) role in β -cell repolarization. We proved that BBR directly bound the KCNH6 protein in pancreatic β -cells (Fig. 2) and inhibited KCNH6 currents (Fig. 4). The effects of BBR were abolished in *Kcnh6* global KO and β KO mice. Thus, BBR specifically targeted the KCNH6 channel in pancreatic β -cells. We have added this information to the manuscript (Page 5, Lines 1 - 7 and Page 14, Lines 11 -13)

25. Hu, H., et al. A potent antiarrhythmic drug N-methyl berbamine extends the action potential through inhibiting both calcium and potassium currents. *Journal of pharmacological sciences* 142, 131-139 (2020).
26. Chen, H., et al. Berberine attenuates spontaneous action potentials in sinoatrial node cells and the currents of human HCN4 channels expressed in *Xenopus laevis* oocytes. *Molecular medicine reports* 10, 1576-1582 (2014).
27. Kim, H.J., Kim, H., Jung, M.H., Kwon, Y.K. & Kim, B.J. Berberine induces pacemaker potential inhibition via cGMP-dependent ATP-sensitive K⁺ channels by stimulating mu/delta opioid receptors in cultured interstitial cells of Cajal from mouse small intestine. *Molecular medicine reports* 14, 3985-3991 (2016).
28. Yu, D., et al. Inhibitory effects and mechanism of dihydroberberine on hERG channels expressed in HEK293 cells. *PLoS one* 12, e0181823 (2017).

Reviewer #2 (Remarks to the Author):

Zhao et.al. in the manuscript entitled "Berberine, a Chinese herb-derived compound, as a novel insulin secretagogue targeting KCNH6 potassium channel" claimed that Berberine was a glucose dependent insulin secretagogue mediated by its effect on closing Kv channel encoded by Kcnh6. The complicated role of KCNH6 in regulating beta cell biology and hence glycemic regulations has been reported by the authors' previous study. It is new to know that KCNH6 might mediate the effect of BBR in beta cells which promotes insulin secretion. I think there are some serious problems in this study to qualify its publication in Nature communications.

First, there are several issues in the animal study which was lousy and ill-designed.

In the Figure2, the insulin releasing test curve of HFD fed mice was uncommonly low and flat, according to our experience and ponderous published evidence.

Thus, I am doubting if the HFD model is even successful, there are no data in this paper to verify this. And I am astonished that the author used global KO mice in HFD model and conditional KO mice in chow diet model, the metabolic phenotype compared to control in neither of the mouse models has been previously reported or described in the current study. The rationale under this kind of design is out of my knowledge.

Further, as to the Kcnh6KO mice, the authors did not offer the information of the animal age, according to the authors' own study that different age of Kcnh6KO mice have opposite metabolic phenotype. The dysfunction of KCNH6 causes overstimulation of insulin secretion in the short term and β cell failure in the long term. So, in which period the authors intended to test the effect of BBR might affect the phenotype. Furthermore, the previous studies of the authors stated that the mechanism underlying the hyperglycemia Kcnh6KO mice is mainly relied on the increased beta cell ER stress and apoptosis, it would not be surprise that the dysfunctional beta cell could not respond to a weak secretagogue as BBR.

Answer: Thank you very much for your comments.

First, we apologize for our carelessness and failure to present the metabolic data from the HFD model; we greatly appreciate you for bringing this omission to our

attention. In this study, the mice were fed the HFD from 4 to 12 weeks and the increased body weight of HFD-fed mice confirmed the successful establishment of the model (new Supplementary Fig. S1a). The metabolic phenotype and detailed information of the HFD-fed mice are provided in new Supplementary Fig. S1b-g.

Second, we agree that HFD-fed mice should exhibit increased insulin secretion. However, our insulin secretion result is not an exception¹⁻³. For example, the previously published paper (*Diabetes 2011*) also showed a flat plasma insulin level (the figure is presented below) after glucose stimulation under HFD conditions, similar to our results.

Akerfeldt, MC, et al. Inhibition of Id1 augments insulin secretion and protects against high-fat diet-induced glucose intolerance. *Diabetes*, 2011; 60(10): 2506-14. doi:10.2337/db11-0083

Third, you asked why we used global KO mice in the HFD model and β KO mice in the chow diet model. This question is very good and would puzzle most scientists; we should have explained it. According to our findings, the global KO mice did not exhibit impaired glucose tolerance until at least 12-14 weeks old⁴ when fed a normal chow diet, whereas the β KO mice exhibited impaired glucose tolerance as early as 8 weeks old when fed a normal chow diet (Supplementary Fig. S4). **Therefore, the disease phenotype in animals with cell-specific gene KO may be more obvious than in mice with a global KO.** The detailed mechanism underlying the phenotypic discrepancies was unclear, but recent studies assessing **gene compensation**^{5,6} may at least provide an explanation; however, the exact mechanism still needs to be investigated. As the aim of our study is to examine the insulinotropic effect of BBR on mice with impaired glucose tolerance and relatively sufficient β -cell function, we performed the experiments at the early adult age to avoid serious β -cell failure. Therefore, we used the HFD to induce impaired glucose tolerance in global KO mice. Because the β KO mice exhibited impaired glucose tolerance earlier than global KO

mice, we did not feed them the HFD and performed the experiment at the age of 8 weeks. Meanwhile, we used high-dose (3 g/kg bw) glucose loading in the normoglycemic mice as controls for the β KO mice.

Finally, you mentioned our previous studies showing that the mechanism underlying the hyperglycemia in *Kcnh6* KO mice mainly relied on the increased β -cell ER stress and apoptosis, and thus the observation that the dysfunctional β -cells did not respond to a weak secretagogue such as BBR was not surprising. However, the *KCNH6* gene mutation is different from the effect of the BBR treatment on the KCNH6 channel. Mutation will lead to permanent dysfunction of the KCNH6 channel and explains why old *Kcnh6* KO mice showed impaired insulin secretion due to ER stress and apoptosis in β -cells. However, the effect of BBR is temporary and it only works under high-glucose conditions to relieve hyperglycemia by increasing insulin secretion. As shown in our previous papers, older KO mice exhibited obvious β -cell ER stress and apoptosis^{4,7}, and thus we chose younger KO mice (before 14 wks) to compare the effect of BBR on insulin secretion in WT and KO mice without β -cell ER stress and apoptosis.

We have added this explanation to the manuscript (Page 6, Lines 10 - 24).

1. Akerfeldt, M.C. & Laybutt, D.R. Inhibition of *Id1* augments insulin secretion and protects against high-fat diet-induced glucose intolerance. *Diabetes* 60, 2506-2514 (2011).
2. Eto, K., et al. Genetic manipulations of fatty acid metabolism in beta-cells are associated with dysregulated insulin secretion. *Diabetes* 51 Suppl 3, S414-420 (2002).
3. Dezaki, K., et al. Blockade of pancreatic islet-derived ghrelin enhances insulin secretion to prevent high-fat diet-induced glucose intolerance. *Diabetes* 55, 3486-3493 (2006).
4. Yang, J.K., et al. From Hyper- to Hypoinsulinemia and Diabetes: Effect of KCNH6 on Insulin Secretion. *Cell Rep* 25, 3800-3810 e3806 (2018).
5. El-Brolosy, M.A., et al. Genetic compensation triggered by mutant mRNA degradation. *Nature* 568, 193-197 (2019).
6. Ma, Z., et al. PTC-bearing mRNA elicits a genetic compensation response via *Upf3a* and *COMPASS* components. *Nature* 568, 259-263 (2019).
7. Lu, J., et al. KCNH6 protects pancreatic beta-cells from endoplasmic reticulum stress and apoptosis. *FASEB J* 34, 15015-15028 (2020).

Second, how the BBR was dissolved, where it was purchased from or which kind of reagent they used as placebo in human subjects were not clarified in the manuscript.

Answer: For the animal studies and *in vitro* studies, BBR was purchased from Sigma-Aldrich (PHR1502, Lot#LRAA9232) and dissolved in DMSO. For the human study, we obtained tablets of BBR and the placebo from Xin Yi Tian Ping

Pharmaceutical Co., Ltd (Shanghai, China). Tablets of BBR (China Pharmacopoeia, Volume 2, 2015 Edition) manufactured by Xin Yi Tian Ping Pharmaceutical Co., Ltd. were approved for marketing by the State Food and Drug administration (SFDA) of China. Certificates of analysis of the products are provided below. We added this information to the Methods section in the main text (Page 16, Lines 5 - 10).

English translation of the *Placebo* inspection report

Shanghai XinYiTianPing Pharmaceutical Co., Ltd.

Finished product inspection report

Report No. 001

上海信谊天平药业有限公司
上海信谊天平药业有限公司
成品检验报告书

报告书编号: 001

品名/规格	盐酸小檗碱片0.1g	生产日期	2018年04月14日
包装规格	100片/瓶	有效期至	2020年04月13日
批号	180401	报告日期	2018年05月11日
批数量	万片	检验依据	中国药典 2015 版二部

检验项目	内控规定	检验结果	结论
性状	本品为黄色片、糖衣片或薄膜衣片，除去包衣后显黄色	本品为薄膜衣片，除去薄膜衣后显黄色，外观符合规定	合格
鉴别	A. 即产生沉淀 B. 即显樱红色 C. 滤液显氯化物鉴别(1)的反应。	-----	-----
检查	溶出度	≥70%	-----
	重量差异限度	±7.5%	-----
	含量限度	93.0%-107.0%	-----
微生物限度	需氧菌总数	≤10 ⁵ cfu/g	<10 cfu/g 合格
	霉菌和酵母菌总数	≤10 ² cfu/g	<10 cfu/g 合格
	大肠埃希菌	不得检出	未检出 合格
检验结论	检验结果符合规定		
检验员: 王爱珍 2018.05.11	复核员: 毛健 2018.05.11	审核人: 张佩 2018.05.11	

Product name/specification	Berberine/0.1 g	Manufacturing date	2018-04-14
Package specification	100 tablets/bottle	Expire date	2020-04-13
Batch number	180401	Report date	2018-05-11
Batch quantity	10 thousand tablets	Inspection basis	China Pharmacopoeia (V2, 2015 Edition)

Items of inspection	Internal control regulations	Test results	Conclusion
Character	This product is yellow tablets, sugar-coated tablets or film-coated tablets, and is yellow after detaching its coating.	This product is film-coated tablets and is yellow after detaching its coating. The appearance meets the requirements.	Qualified
Identification	A. Precipitation occurs. B. The color is cherry-red C. The filtrate shows the reaction of chloride identification (1).	-----	-----
Inspection	Dissolution	≥70%	-----
	Weight variation limit	±7.5%	-----
Content limit	93.0%-107.0%	-----	-----
Microbial limit	Total aerobe	≤10 ⁵ cfu/g	<10 cfu/g Qualified
	Total mycetes and saccharomycetes	≤10 ² cfu/g	<10 cfu/g Qualified
	Escherichia coli	No detection	Not detected Qualified
Test conclusion	The test results meet the requirements.		
Inspector:	Reviewer:	Auditor:	

English translation of the *Berberine* inspection report

Shanghai XinYiTianPing Pharmaceutical Co., Ltd.

Finished product inspection report

Report No. 001

上海信谊天平药业有限公司 成品检验报告书				
报告书编号: 001				
品名/规格	盐酸小檗碱片0.1g	生产日期	2018年05月24日	
包装规格	100片/瓶	有效期至	2020年05月23日	
批号	23180501	报告日期	2018年06月28日	
批数量	7万片	检验依据	中国药典2015版二部	
检验项目	内控规定	检验结果	结论	
性状	本品为黄色片,糖衣片或薄膜衣片,除去包衣后显黄色	本品为薄膜衣片,除去薄膜衣后显黄色,外观符合规定	合格	
鉴别	A.即产生沉淀 B.即显樱红色 C.滤液显氯化物鉴别(1)的反应。	A.即产生沉淀 B.即显樱红色 C.滤液显氯化物鉴别(1)的反应。	合格	
检查	溶出度	≥70%	100.0%	合格
	重量差异限度	±7.5%	+3.5%; -3.1%	合格
	含量限度	93.0%-107.0%	99.4%	合格
微生物限度	需氧菌总数	≤10 ³ cfu/g	<10 cfu/g	合格
	霉菌和酵母菌总数	≤10 ² cfu/g	<10 cfu/g	合格
	大肠埃希菌	不得检出	未检出	合格
检验结论		检验结果符合规定		
检验员:	王亚强 2018.06.28	复核员:	王亚强 2018.06.28	
		审核人:	王亚强 2018.06.28	

Product name/specification	Berberine/0.1 g	Manufacturing date	2018-05-24	
Package specification	100 tablets/bottle	Expire date	2020-05-23	
Batch number	23180501	Report date	2018-06-28	
Batch quantity	10 thousand tablets	Inspection basis	China Pharmacopoeia (V2, 2015 Edition)	
Items of inspection	Internal control regulations	Test results	Conclusion	
Character	This product is yellow tablets, sugar-coated tablets or film-coated tablets, and is yellow after detaching its coating.	This product is film-coated tablets and is yellow after detaching its coating. The appearance meets the requirements.	Qualified	
Identification	A. Precipitation occurs. B. The color is cherry-red C. The filtrate shows the reaction of chloride identification (1).	A. Precipitation occurs. B. The color is cherry-red C. The filtrate shows the reaction of chloride identification (1).	Qualified	
Inspection	Dissolution	≥70%	100%	Qualified
	Weight variation limit	±7.5%	+3.5%; -3.1%	Qualified
Content limit	93.0%-107.0%	99.4%	Qualified	
Microbial limit	Total aerobe	≤10 ³ cfu/g	<10 cfu/g	Qualified
	Total mycetes and saccharomycetes	≤10 ² cfu/g	<10 cfu/g	Qualified
	Escherichia coli	No detection	Not detected	Qualified
Test conclusion	The test results meet the requirements.			
Inspector:	Reviewer:	Auditor:		

For, there are multiple evidence published about the effect of Berberine on regulating the insulin secretion, most of which are opposite to this study.

Published reports have found the potential mechanisms relating to attenuated mitochondria oxidative phosphorylation via stimulating AMPK and inhibiting PKA signaling. The author should discuss and provide data how the BBR they used in this study affected the cell growth, which was the toxic dosage and discuss why the phenotype was opposite with most other studies.

Answer: Thank you for bringing these issues to our attention.

First, we confirmed that the indicated concentrations of BBR exerted a non-toxic effect on β -cell viability (new Fig.1b). We searched PubMed for studies published in English up to February 14, 2021, with the search terms “((BBR) OR (Berberine)) AND (insulin secretion)”, and we found 19 papers related to this topic⁸⁻²⁶. Of these

19 papers, **11** studies documented a positive effect of BBR on promoting insulin secretion^{9,12-14,17-23}. **Four** *in vivo* studies reported decreased insulin levels after long-term BBR treatment due to increased glucose concentrations^{8,10,15,16}. Only **4** papers described negative effects on insulin secretion after the administration of BBR^{11,24-26}. The opposite results might be due to different experimental conditions, especially the lower purity of BBR *in vitro*. As BBR is a natural product purified from plants, the components of this product are extraordinarily complex, and may include some harmful impurities. We noticed that **3** of those **4** papers obtained BBR from National Institute for the Control of Pharmaceutical and Biological Products^{11,25,26}. This institute does not produce any chemicals itself, and we were unable to find any information (constitution and purity) for BBR on the website of this institute (<https://www.nifdc.org.cn/>). Our BBR was purchased from Sigma-Aldrich (PHR1502, Lot#LRAA9232) with a high purity exceeding 99%. We confirmed increased insulin secretion in both mouse islets *in vitro* and humans and mice *in vivo*.

Second, we agree that many papers have reported the potential mechanisms of BBR related to the AMPK and PKA pathways. However, glucose-dependent hypoglycemic drugs such as metformin and GLP-1 analogues, as well as insulin secretagogues, also affect AMPK and PKA pathways²⁷⁻³¹, although none of them target the AMPK or PKA pathways directly. Therefore, we postulated that BBR might affect the AMPK and PKA pathways in β -cells, but this effect might occur after relieving the abnormally high glucose metabolism through increased insulin secretion, similar to most insulin secretagogues or other hypoglycemic drugs. In our study, we focused on investigating the direct effect of BBR on insulin secretion through the inhibition of the KCNH6 channel expressed on islet β -cells from mice and humans. We have added this information to the manuscript (Page 14, Lines 20 - Page 15, Line 2).

8. Cole, L.K., et al. Supplemental Berberine in a High-Fat Diet Reduces Adiposity and Cardiac Dysfunction in Offspring of Mouse Dams with Gestational Diabetes Mellitus. *The Journal of nutrition* (2021).

9. Li, J., et al. Amorphous solid dispersion of Berberine mitigates apoptosis via iPLA2beta/Cardiolipin/Opa1 pathway in db/db mice and in Palmitate-treated MIN6 beta-cells. *International journal of biological sciences* 15, 1533-1545 (2019).

10. Sun, Y., et al. Restoration of GLP-1 secretion by Berberine is associated with protection of colon enterocytes from mitochondrial overheating in diet-induced obese mice. *Nutrition & diabetes* 8, 53 (2018).

11. Bai, M., et al. Berberine inhibits glucose oxidation and insulin secretion in rat islets. *Endocrine journal* 65, 469-477 (2018).
12. Jiang, Y.Y., et al. Protective role of berberine and *Coptischinensis* extract on T2MD rats and associated islet *Rin5f* cells. *Molecular medicine reports* 16, 6981-6991 (2017).
13. Dong, Y., et al. Metabolomics Study of Type 2 Diabetes Mellitus and the AntiDiabetic Effect of Berberine in Zucker Diabetic Fatty Rats Using Uplc-ESI-Hdms. *Phytotherapy research : PTR* 30, 823-828 (2016).
14. Liu, L., et al. Uncoupling protein-2 mediates the protective action of berberine against oxidative stress in rat insulinoma INS-1E cells and in diabetic mouse islets. *British journal of pharmacology* 171, 3246-3254 (2014).
15. Perez-Rubio, K.G., Gonzalez-Ortiz, M., Martinez-Abundis, E., Robles-Cervantes, J.A. & Espinel-Bermudez, M.C. Effect of berberine administration on metabolic syndrome, insulin sensitivity, and insulin secretion. *Metabolic syndrome and related disorders* 11, 366-369 (2013).
16. Shan, C.Y., et al. Alteration of the intestinal barrier and GLP2 secretion in Berberine-treated type 2 diabetic rats. *The Journal of endocrinology* 218, 255-262 (2013).
17. Rayasam, G.V., et al. Identification of berberine as a novel agonist of fatty acid receptor GPR40. *Phytotherapy research : PTR* 24, 1260-1263 (2010).
18. Gao, N., Zhao, T.Y. & Li, X.J. The protective effect of berberine on beta-cell lipooptosis. *Journal of endocrinological investigation* 34, 124-130 (2011).
19. Yu, Y., et al. Modulation of glucagon-like peptide-1 release by berberine: in vivo and in vitro studies. *Biochemical pharmacology* 79, 1000-1006 (2010).
20. Lu, S.S., et al. Berberine promotes glucagon-like peptide-1 (7-36) amide secretion in streptozotocin-induced diabetic rats. *The Journal of endocrinology* 200, 159-165 (2009).
21. Wang, Z.Q., et al. Facilitating effects of berberine on rat pancreatic islets through modulating hepatic nuclear factor 4 alpha expression and glucokinase activity. *World journal of gastroenterology* 14, 6004-6011 (2008).
22. Ko, B.S., et al. Insulin sensitizing and insulinotropic action of berberine from *Cortidis rhizoma*. *Biological & pharmaceutical bulletin* 28, 1431-1437 (2005).
23. Leng, S.H., Lu, F.E. & Xu, L.J. Therapeutic effects of berberine in impaired glucose tolerance rats and its influence on insulin secretion. *Acta pharmacologica Sinica* 25, 496-502 (2004).
24. Lamontagne, J., et al. Pioglitazone acutely reduces insulin secretion and causes metabolic deceleration of the pancreatic beta-cell at submaximal glucose concentrations. *Endocrinology* 150, 3465-3474 (2009).
25. Zhou, L., et al. Berberine acutely inhibits insulin secretion from beta-cells through 3',5'-cyclic adenosine 5'-monophosphate signaling pathway. *Endocrinology* 149, 4510-4518 (2008).
26. Yin, J., et al. Effects of berberine on glucose metabolism in vitro. *Metabolism: clinical and experimental* 51, 1439-1443 (2002).
27. Foretz, M., Guigas, B. & Viollet, B. Understanding the glucoregulatory mechanisms of metformin in type 2 diabetes mellitus. *Nature reviews. Endocrinology* 15, 569-589 (2019).
28. Beiroa, D., et al. GLP-1 agonism stimulates brown adipose tissue thermogenesis and browning through hypothalamic AMPK. *Diabetes* 63, 3346-3358 (2014).
29. Lee, K.Y., Kim, J.R. & Choi, H.C. Gliclazide, a KATP channel blocker, inhibits vascular smooth muscle cell proliferation through the CaMKKbeta-AMPK pathway. *Vascular pharmacology* 102, 21-28 (2018).
30. Wang, Q., Heimberg, H., Pipeleers, D. & Ling, Z. Glibenclamide activates translation in rat pancreatic beta cells through calcium-dependent mTOR, PKA and MEK signalling pathways. *Diabetologia* 51, 1202-1212 (2008).
31. Le, Y., et al. Liraglutide ameliorates palmitate-induced oxidative injury in islet microvascular endothelial cells through GLP-1 receptor/PKA and GTPCH1/eNOS signaling pathways. *Peptides* 124, 170212 (2020).

Third, in Figure 5 b,c, the authors tried to show the changes of Kcnh6 localization in cell membrane, however the authors used the beta-actin as the loading control of the membrane portion, which is indicating the cell plasma not membrane portion.

Answer: We honestly thank you for bringing this issue to our attention. Previously, we did not consider selecting the best loading control. We used beta-actin as the loading control for membrane proteins based on a previous publication. Some papers showed that beta-actin is also present in the plasma membrane fraction or at least associated with plasma membrane (see figure below). We agree that beta-actin is not the best marker for the plasma membrane fraction. According to your suggestion, we

replaced beta-actin with Na-K-ATPase (a classic marker of plasma membrane) as the loading control for the plasma membrane fraction (new Supplementary Fig. S8).

Rangel R, Dobroff AS, Guzman-Rojas L, et al. Targeting mammalian organelles with internalizing phage (iPhage) libraries. *Nat Protoc.* 2013;8(10):1916-1939.

Fourth, in Figure 6, details of the methodology for the clamp study were not explained clearly. For instance, Line 505-506, the authors just claimed that “--- insulin and proinsulin C-peptide were obtained at intervals throughout the clamp study”, but at what time interval, 5 minutes or longer? It is not mentioned in the Method or any other places in the manuscript. It seems to me that the time interval for the whole clamp study was all the same throughout the procedure. However, for most of the hyperglycemic clamp studies, the time intervals for insulin obtainment should be different between the first and second phase. Please clarify. And I also doubted the result of 6c-j. The differences between the calculated AUC value was much smaller than what have been presented in the insulin or C peptide curve between the treat and placebo group. I guess the significant differences might be resulted by an outlier presented in the scatter plots in Figure 6d-f or Figure 6h-j. Most importantly, in all the experiments the authors conducted, BBR was used with a single dose, which is fine to observe in the cell electrophysiology changes or acute effect on insulin secretion, however, a single dosage before IPGTT, a single dosage before hyperglycemic clamp? I did not see the relevance of this acute effect of BBR in regulating beta cell biology as well as long term glucose homeostasis.

Answer: Thank you very much for your careful review. The insulin and C-peptide levels were measured every 2 min within the first 14 min and every 10 min afterwards. We have clarified this procedure in the Methods and refined the details of

methodology for the clamp study (Page 23, Lines 25 – Page 24, Line 3). During the experimental procedure, volunteer No. 14 exhibited higher insulin and C-peptide secretion than other volunteers. We excluded his data and recalculated the *P* value; the significant differences were even larger, as the *P* value changed from 0.0235 to 0.011 and from 0.0071 to 0.0014 for total insulin and C-peptide iAUCs, respectively (see the figures below). Since individual differences in insulin secretion³² and drug susceptibility^{33,34} occur, we propose that we should retain these data.

The protocol for the hyperglycemic clamp study was based on a previous publication³². Studies of the effects of insulin secretagogues on the GTT and hyperglycemic clamp usually used a single dose³⁵⁻³⁸.

32. DeFronzo, R.A., Tobin, J.D. & Andres, R. Glucose clamp technique: a method for quantifying insulin secretion and resistance. *Am J Physiol* 237, E214-223 (1979).

33. Jones, A.G., et al. Markers of beta-Cell Failure Predict Poor Glycemic Response to GLP-1 Receptor Agonist Therapy in Type 2 Diabetes. *Diabetes Care* 39, 250-257 (2016).

34. Sattiraju, S., Reyes, S., Kane, G.C. & Terzic, A. K(ATP) channel pharmacogenomics: from bench to bedside. *Clinical pharmacology and therapeutics* 83, 354-357 (2008).

35. Ligtenberg, J.J., Venker, C.E., Sluiter, W.J., Reitsma, W.D. & Van Haefken, T.W. Effect of glibenclamide on insulin release at moderate and high blood glucose levels in normal man. *Eur J Clin Invest* 27, 685-689 (1997).

36. Eldor, R., et al. Discordance Between Central (Brain) and Pancreatic Action of Exenatide in Lean and Obese Subjects. *Diabetes Care* 39, 1804-1810 (2016).

37. Gjesing, A.P., et al. High heritability and genetic correlation of intravenous glucose- and tolbutamide-induced insulin secretion among non-diabetic family members of type 2 diabetic patients. *Diabetologia* 57, 1173-1181 (2014).

38. Hameed, A., et al. Coixol amplifies glucose-stimulated insulin secretion via cAMP mediated signaling pathway. *European journal of pharmacology* 858, 172514 (2019).

Last but not the least, I do not have question about the electrophysiology study the author conducted in BBR treated and *Kcnh6KO* cells. If the authors linked this alteration to the insulin secretion, I am wondering whether glucose induced membrane capacitance alteration measured by whole cell patch clamp was affected by BBR or loss of *Kcnh6KO*, which monitors bona fide insulin granule exocytosis.

Answer: Thank you very much for your suggestion. We understand that the β -cell membrane capacitance will increase during insulin granule exocytosis, so measuring membrane capacitance can directly monitor insulin secretion. However, this technique may not be suitable to monitor the effects of voltage-dependent channel blockers such as BBR, because the membrane potential must be fixed under the voltage-clamp in LockIn mode to record the membrane capacitance³⁹⁻⁴¹. As BBR inhibited voltage-dependent potassium channel to promote insulin secretion, BBR would be useless if the membrane potential was artificially fixed. We performed the experiment in primary cultured pancreatic β -cells from WT mice and found that BBR failed to

affect the membrane capacitance changes compared to control group (see figures below). This result was also consistent with the finding from a previous study that tolbutamide and diazoxide do not affect the changes in β -cell capacitance⁴².

Nevertheless, we have added new data (new Fig. 5j-l) from the direct analysis of the effect of BBR on insulin exocytosis. We performed a total internal reflection fluorescent (TIRF) microscopy analysis of insulin granules labeled with insulin-EGFP (Fig. 5j-l). The total number of insulin exocytosis events after BBR treatment was considerably increased compared with those of vehicle-treated cells (Fig. 5k-l). Based on these results, BBR directly promoted insulin granule exocytosis induced by high glucose. Furthermore, we directly examined the ability of BBR to induce insulin secretion and modulate glucose-induced increases in intracellular Ca^{2+} levels (Supplementary Fig. S5). Therefore, we propose that the BBR-induced electrophysiological alteration will increase calcium influx and then subsequently increase insulin secretion (Page 11, Lines 10 - 15).

39. Neef, A., Heinemann, C. & Moser, T. Measurements of membrane patch capacitance using a software-based lock-in system. *Pflügers Archiv : European journal of physiology* 454, 335-344 (2007).

40. Kang, L., et al. *Munc13-1 is required for the sustained release of insulin from pancreatic beta cells. Cell metabolism* 3, 463-468 (2006).

41. Pigeau, G.M., et al. *Insulin granule recruitment and exocytosis is dependent on p110gamma in insulinoma and human beta-cells. Diabetes* 58, 2084-2092 (2009).

42. Mariot, P., Gilon, P., Nenquin, M. & Henquin, J.C. *Tolbutamide and diazoxide influence insulin secretion by changing the concentration but not the action of cytoplasmic Ca^{2+} in beta-cells. Diabetes* 47, 365-373 (1998).

It is not new that BBR exerts broad effects on metabolic organs such as liver, the adipose, endocrine pancreas and muscle tissue, even in gut microbiota. I am not convinced by the authors that the effect of BBR on Kv Channel in beta cell was linked to the phenotype of mice glucose excursion and beta cell secretion. Several gaps need

to be filled and scientific significance and novelty of this study were also questionable.

Answer: We appreciate your suggestion that as a natural compound, BBR may exert broad effects on metabolic organs in addition to pancreatic β -cells. Even a chemical compound with one target may act on different organs and have different functions. For example, sulfonylureas stimulate insulin secretion by blocking the $K_{ATP}/SUR1$ channel in islet β -cells, while these compounds also act on the $K_{ATP}/SUR2$ channel in cardiac cells. In this project, we focused on the effect of BBR on insulin secretion. Both the analysis of glucose-stimulated insulin secretion from mouse islets and hyperglycemic clamp study in humans indicated that main function of BBR in pancreatic β -cell is to promote insulin secretion in a glucose dependent-manner. In the future, we will continue to examine the effects of KCNH6 and BBR on other metabolic organs.

Reviewers' Comments:

Reviewer #1:

Remarks to the Author:

The authors have made a very comprehensive effort to address the previous concerns. I have only one more comment: I think the title should be changed to "Berberine is a novel insulin secretagogue targeting the KCNH6 potassium channel". As the authors point out, when berberine is actually derived from Chinese herbs there can be many contaminants. The authors are specifically analyzing the effects of the pure compound.

Reviewer #3:

Remarks to the Author:

This is a very well conducted experimental study by the group, likely to be motivated by their report of a KCNH6 mutation affecting a multigenerational family (Yang JK, 2018) with hypo and hyperglycaemia for which they have generated KCNH6 global knock-out (ko) and beta cell-knock out (bko) mice which demonstrated the dysregulation of insulin secretion leading to ER stress and cellular apoptosis.

In this paper, they explored the glucose lowering mechanism of berberine (BBR), a Chinese herb widely used for treating infections, on lowering glucose in a series of cellular, animal and human studies. The premise of the study relates to the effects of BBR on the activity of the non-ATP dependent but voltage-dependent potassium (Kv) channels resulting in closure of the voltage-dependent calcium channels (VDCC) which are key events in the release of insulin from beta cells .

The author sourced BBR as a single molecule from the industry and tested its toxicity on viability of beta cells in different doses. They treated HFD-wild type, HFD-ko and chow-diet fed bko mice with high dose and low dose BBR and demonstrated the glucose lowering and insulin and C-peptide elevating effects of BBR.

In cell-based and islet perfusion experiments, they demonstrated the glucose lowering effects of BBR in high glucose but not low glucose concentration medium. They also demonstrated that these effects were not influenced by the addition of tolbutamide which target the ATP dependent K channel but negated by silencing the KCNH6 gene or treatment with verapamil that target the same gene.

They went on and performed the channel current study and reported that BBR accelerated the closure of the Kv channels and that BBR increased the expression of KCNH6 on the cellular members of insulin secreting cells with stimulation of exocytosis and release of insulin.

They then performed a hyperglycemic clamp study in 15 healthy volunteers and confirmed that BBR increased the insulin and CP concentration compared with placebo when the BG was clamped at 12 mmol/L for 160 minutes.

Discovery of disease-associated genetic mutations in human can lead to discovery of novel biology which may be therapeutic targets and ion channels may well be one such pathway, as in the case of SU/glinides. This novel study was motivated by the authors' report of a MODY family which has provided clinical relevance to these results. This point should be mentioned earlier in the introduction to give readers a better context.

However, KCHN6 gene is also the same gene that has been implicated in prolonged QT interval which may lead to arrhythmia as a rare syndrome. Thus, efficacy aside, it is of utmost importance that the cardiac toxicity of BBR must be explored before we claim the clinical utility of BBR. Indeed, cardiac toxicity due to co-administration of BBR and macrolides that share the same cytochrome family for metabolism has been reported (Zhi D et al, 2015). In this light, what is the cardiac phenotype of the global ko and beta-cell ko mice and what were their survival rates compared to the wild types? In this version, the authors have clarified many albeit not all experimental details and performed additional experiments to address the reviewer's concerns.

Compared to the extensive report on the experimental studies, the details of the phase 1 PD study of BBR in human volunteers was minimal. Several important points such as ethics approval and the PK parameters of BBR are not stated. Only single dose study was performed and on what basis was the human dose of BBR selected? How was the selected dose of BBR compared to the dose used in clinical practice?

The authors only performed hyperglycemic clamp and it is not appropriate to claim that BBR did not stimulate insulin secretion during hypoglycemic condition for which specific experiment had not been designed or conducted. During the conduct of the phase 1 study, were the participants monitored for any ECG changes since changes in K/Ca trafficking and action potential may induce ECG changes. Did the participants experience any side effects and was there any delayed hypoglycaemia after the clamp study?

Response to Reviewers

Reviewer #1 (Remarks to the Author):

The authors have made a very comprehensive effort to address the previous concerns. I have only one more comment: I think the title should be changed to "Berberine is a novel insulin secretagogue targeting the KCNH6 potassium channel". As the authors point out, when berberine is actually derived from Chinese herbs there can be many contaminants. The authors are specifically analyzing the effects of the pure compound.

Answer: Thank you very much for your work on reviewing our manuscript. We agree with your comment, and we have changed the title to "Berberine is a novel insulin secretagogue targeting the KCNH6 potassium channel". (Page 1, Line 1-2)

Reviewer #3 (Remarks to the Author):

This is a very well conducted experimental study by the group, likely to be motivated by their report of a KCNH6 mutation affecting a multigenerational family (Yang JK, 2018) with hypo and hyperglycaemia for which they have generated KCNH6 global knock-out (ko) and beta cell-knock out (bko) mice which demonstrated the dysregulation of insulin secretion leading to ER stress and cellular apoptosis.

In this paper, they explored the glucose lowering mechanism of berberine (BBR), a Chinese herb widely used for treating infections, on lowering glucose in a series of cellular, animal and human studies. The premise of the study relates to the effects of BBR on the activity of the non-ATP dependent but voltage-dependent potassium (Kv) channels resulting in closure of the voltage-dependent calcium channels (VDCC) which are key events in the release of insulin from beta cells.

The author sourced BBR as a single molecule from the industry and tested its toxicity on viability of beta cells in different doses. They treated HFD-wild type, HFD-ko and chow-diet fed bko mice with high dose and low dose BBR and demonstrated the glucose lowering and insulin and C-peptide elevating effects of BBR.

In cell-based and islet perfusion experiments, they demonstrated the glucose lowering effects of BBR in high glucose but not low glucose concentration medium. They also demonstrated that these effects were not influenced by the addition of tolbutamide which target the ATP dependent K channel but negated by silencing the KCNH6 gene or treatment with verapamil that target the same gene.

They went on and performed the channel current study and reported that BBR accelerated the closure of the Kv channels and that BBR increased the expression of KCNH6 on the cellular members of insulin secreting cells with stimulation of exocytosis and release of insulin.

They then performed a hyperglycemic clamp study in 15 healthy volunteers and confirmed that BBR increased the insulin and CP concentration compared with placebo when the BG was clamped at 12 mmol/L for 160 minutes.

Discovery of disease-associated genetic mutations in human can lead to discovery of novel biology which may be therapeutic targets and ion channels may well be one such pathway, as in the case of SU/glinides. This novel study was motivated by the authors' report of a MODY family which has provided clinical relevance to these results. This point should be mentioned earlier in the introduction to give readers a better context.

Answer: Thank you very much for your comment. We highly agree with your opinion that discovery of disease-associated genetic mutations in human can lead to discovery of novel biology which may be therapeutic targets and ion channels may well be one

such pathway, as in the case of SU/glinides. Indeed, this study was motivated by our previous report of a MODY family with KCNH6 gene mutation ¹, we therefore discovered KCNH6 potassium as the therapeutic target to treat diabetes. We have added this point in the introduction (Page 3, Line 21 – Page 4, Line 1), described as followed:

Discovery of disease-associated genetic mutations in human can lead to discovery of novel biology which may be therapeutic targets, and ion channels may well be one such pathway, as in the case of sulfonylureas and glinides ^{2,3}. Recently, we identified the pivotal role of KCNH6, a voltage-dependent K⁺ (K_v) channel, in insulin secretion, by studying a large four-generation family with monogenic diabetes ¹. GSIS from pancreatic islet β-cells is influenced by high glucose-dependent repolarization caused by K_v channels such as KCNH6 ⁴. In this study, we linked BBR with the KCNH6 protein at the molecular level and elucidated the specific mechanism by which BBR stimulates insulin secretion. Blockade of KCNH6 channels by BBR increases insulin secretion in a *high glucose-dependent manner* or only under hyperglycemic conditions, suggesting that BBR is a new glucose-dependent insulin secretagogue for the treatment of diabetes that does not cause hypoglycemia.

References:

- 1 Yang, J. K. et al. From Hyper- to Hypoinsulinemia and Diabetes: Effect of KCNH6 on Insulin Secretion. Cell reports 25, 3800-3810 e3806, doi:10.1016/j.celrep.2018.12.005 (2018).
- 2 Hugill, A., Shimomura, K., Ashcroft, F. M. & Cox, R. D. A mutation in KCNJ11 causing human hyperinsulinism (Y12X) results in a glucose-intolerant phenotype in the mouse. Diabetologia 53, 2352-2356, doi:10.1007/s00125-010-1866-x (2010).
- 3 Huopio, H. et al. A new subtype of autosomal dominant diabetes attributable to a mutation in the gene for sulfonylurea receptor 1. Lancet 361, 301-307, doi:10.1016/S0140-6736(03)12325-2 (2003).
- 4 Herrington, J. et al. Blockers of the delayed-rectifier potassium current in pancreatic beta-cells enhance glucose-dependent insulin secretion. Diabetes 55, 1034-1042, doi:10.2337/diabetes.55.04.06.db05-0788 (2006).

However, KCHN6 gene is also the same gene that has been implicated in prolonged QT interval which may lead to arrhythmia as a rare syndrome. Thus, efficacy aside, it is of utmost importance that the cardiac toxicity of BBR must be explored before we claim the clinical utility of BBR. Indeed, cardiac toxicity due to co-administration of BBR and macrolides that share the same cytochrome family for metabolism has been reported (Zhi D et al, 2015). In this light, what is the cardiac phenotype of the global ko and beta-cell ko mice and what were their survival rates compared to the wild types? In this version, the authors have clarified many albeit not all experimental details and performed additional experiments to address the reviewer's concerns.

Answer: Thank you for your comment. This is a very good question. KCNH6 (encoding K_v11.2) gene is in the same KCNH family of another gene, KCNH2 (HERG, encoding K_v11.1), and KCNH2 (HERG) is known to induce the Long QT syndrome ⁵. We found that KCNH6 was dominant in pancreatic islets while KCNH2 was dominant in heart (data are shown as below).

This was also consistent with another group's result indicating that the expression of *KCNH6* was higher than *KCNH2* in pancreatic islets (see figure below). These findings suggest that *KCNH2* mainly function in heart to induce the long QT syndrome while *KCNH6* mainly function in islet β -cells to promote insulin secretion.

Hardy AB, Fox JE, Giglou PR, Wijesekara N, Bhattacharjee A, Sultan S, Gyulkhandanyan AV, Gaisano HY, MacDonald PE, Wheeler MB. Characterization of Erg K⁺ channels in alpha- and beta-cells of mouse and human islets. *J Biol Chem*. 2009 Oct 30;284(44):30441-52.

We performed the ECG analysis of global KO and β KO mice (new Supplementary Fig. 5) and of WT littermates from the same breeding pair as controls. We did not find any difference compared to the controls. Further, we did not observe any difference in survival rates at the time of more than 1 year age of global KO and β KO mice compared to WT mice.

We have added such description in the manuscript as followed (Page 6, Line 15-18):

KCNH6 gene is in the same KCNH family of another gene, *KCNH2* (*HERG*), and *HERG* is known to induce the Long QT syndrome⁵. We, thus, examined the cardiac phenotype of both global KO mice and β KO mice. No difference was observed in electrocardiogram (ECG) results compared to the control mice (Supplementary Fig. 5).

Supplementary Fig. 5: The electrocardiogram (ECG) analysis of global KO and β KO mice

a, Representative ECG recorded from WT and global KO mice, and from control and β KO mice.

b, ECG measurements of the WT and global KO mice, and the control and β KO mice. Statistical significance was assessed using Student's *t* test (two-sided).

Reference:

5 Zhou, Z., Gong, Q., Epstein, M. L. & January, C. T. HERG channel dysfunction in human long QT syndrome. Intracellular transport and functional defects. *J Biol Chem* 273, 21061-21066, doi:10.1074/jbc.273.33.21061 (1998).

Compared to the extensive report on the experimental studies, the details of the phase I PD study of BBR in human volunteers was minimal. Several important points such as ethics approval and the PK parameters of BBR are not stated. Only single dose study was performed and on what basis was the human dose of BBR selected? How was the selected dose of BBR compared to the dose used in clinical practice?

Answer: We honestly thank you for bringing this omission to our attention. First, the ethics approval and its translation, a completed CONSORT checklist and a copy of the study protocol and its translation as approved by our institutional review board are presented in this version (uploaded as **new** Supplementary Info Files). Second, previous literatures studying the PK parameters of BBR showed that after single oral dose of BBR, the $t_{1/2\alpha}$ was 0.869 - 0.87 h, t_{max} was 2.37 - 4.0 h⁶⁻⁸. Third, a meta-analysis of 27 clinical studies and 2569 participants has evaluated BBR at dosages ranging from 600 mg to 2700 mg per day for as long as six months in adults⁹.

The most common and most effective dosage appears to be 1500 mg daily. There is insufficient evidence to support its use in children.

We selected 1000 mg as the moderate dose of BBR and performed the hyperglycemic clamp study 1 h after BBR administration to reach a relatively high blood concentration of BBR. Studies that evaluating the effects of insulin secretagogues usually used a single dose of the drug¹⁰⁻¹³. Besides, this study mainly focused on the transient effect of BBR on insulin secretion, therefore, we chose to give BBR as single dose.

We have added such explanation in the manuscript as followed (Page 12, Line 2-9):

Previous literatures studying the PK parameters of BBR showed that after single oral dose of BBR, the $t_{1/2\alpha}$ was 0.869 - 0.87 h, t_{max} was 2.37 - 4.0 h⁶⁻⁸. Previous clinical studies have evaluated BBR at dosages ranging from 600 mg to 2700 mg per day for as long as six months in adults^{9,14-16}. The most common and most effective dosage appears to be 500 mg three times a day. As studies that evaluating the effects of insulin secretagogues usually used a single dose of the drug¹⁰⁻¹³, a single dose of 1 g of BBR tablets or placebo tablets was administered orally. Then, a 160-min hyperglycemic clamp was performed 1 hour after BBR administration at a baseline blood glucose level of +6.9 mmol/L as the target level of hyperglycemia¹⁷.

References:

- 1 Yang, J. K. et al. From Hyper- to Hypoinsulinemia and Diabetes: Effect of KCNH6 on Insulin Secretion. *Cell reports* 25, 3800-3810 e3806, doi:10.1016/j.celrep.2018.12.005 (2018).
- 2 Hugill, A., Shimomura, K., Ashcroft, F. M. & Cox, R. D. A mutation in KCNJ11 causing human hyperinsulinism (Y12X) results in a glucose-intolerant phenotype in the mouse. *Diabetologia* 53, 2352-2356, doi:10.1007/s00125-010-1866-x (2010).
- 3 Huopio, H. et al. A new subtype of autosomal dominant diabetes attributable to a mutation in the gene for sulfonylurea receptor 1. *Lancet* 361, 301-307, doi:10.1016/S0140-6736(03)12325-2 (2003).
- 4 Herrington, J. et al. Blockers of the delayed-rectifier potassium current in pancreatic beta-cells enhance glucose-dependent insulin secretion. *Diabetes* 55, 1034-1042, doi:10.2337/diabetes.55.04.06.db05-0788 (2006).
- 5 Zhou, Z., Gong, Q., Epstein, M. L. & January, C. T. HERG channel dysfunction in human long QT syndrome. Intracellular transport and functional defects. *J Biol Chem* 273, 21061-21066, doi:10.1074/jbc.273.33.21061 (1998).
- 6 Ye, M., Fu, S., Pi, R. & He, F. Neuropharmacological and pharmacokinetic properties of berberine: a review of recent research. *The Journal of pharmacy and pharmacology* 61, 831-837, doi:10.1211/jpp/61.07.0001 (2009).
- 7 Li, G., Zhao, M., Qiu, F., Sun, Y. & Zhao, L. Pharmacokinetic interactions and tolerability of berberine chloride with simvastatin and fenofibrate: an open-label, randomized, parallel study in healthy Chinese subjects. *Drug design, development and therapy* 13, 129-139, doi:10.2147/DDDT.S185487 (2019).
- 8 Alolga, R. N. et al. Significant pharmacokinetic differences of berberine are attributable to variations in gut microbiota between Africans and Chinese. *Scientific reports* 6, 27671, doi:10.1038/srep27671 (2016).
- 9 Lan, J. et al. Meta-analysis of the effect and safety of berberine in the treatment of type 2 diabetes mellitus, hyperlipemia and hypertension. *Journal of ethnopharmacology* 161, 69-81, doi:10.1016/j.jep.2014.09.049 (2015).
- 10 Ligtgenberg, J. J., Venker, C. E., Sluiter, W. J., Reitsma, W. D. & Van Haeften, T. W. Effect of glibenclamide on insulin release at moderate and high blood glucose levels in normal man. *Eur J Clin Invest* 27, 685-689, doi:10.1046/j.1365-2362.1997.1710716.x (1997).
- 11 Eldor, R. et al. Discordance Between Central (Brain) and Pancreatic Action of Exenatide in Lean and Obese Subjects. *Diabetes Care* 39, 1804-1810, doi:10.2337/dc15-2706 (2016).
- 12 Gjesing, A. P. et al. High heritability and genetic correlation of intravenous glucose- and tolbutamide-induced insulin secretion among non-diabetic family members of type 2 diabetic patients. *Diabetologia* 57, 1173-1181, doi:10.1007/s00125-014-3207-y (2014).
- 13 Hameed, A. et al. Coixol amplifies glucose-stimulated insulin secretion via cAMP mediated signaling pathway. *European journal of pharmacology* 858, 172514, doi:10.1016/j.ejphar.2019.172514 (2019).
- 14 Zhang, Y. et al. Treatment of type 2 diabetes and dyslipidemia with the natural plant alkaloid berberine. *J Clin Endocrinol Metab* 93, 2559-2565, doi:10.1210/jc.2007-2404 (2008).
- 15 Dong, H., Wang, N., Zhao, L. & Lu, F. Berberine in the treatment of type 2 diabetes mellitus: a systemic review and meta-analysis. *Evidence-based complementary and alternative medicine : eCAM* 2012, 591654, doi:10.1155/2012/591654 (2012).
- 16 Yin, J., Xing, H. & Ye, J. Efficacy of berberine in patients with type 2 diabetes mellitus. *Metabolism: clinical and experimental* 57, 712-717, doi:10.1016/j.metabol.2008.01.013 (2008).

The authors only performed hyperglycemic clamp and it is not appropriate to claim that BBR did not stimulate insulin secretion during hypoglycemic condition for which specific experiment had not been designed or conducted. During the conduct of the phase 1 study, were the participants monitored for any ECG changes since changes in K/Ca trafficking and action potential may induce ECG changes. Did the participants experience any side effects and was there any delayed hypoglycaemia after the clamp study?

Answer: Thanks for your comment. First, we measured the fasting (before intravenously glucose injection) glucose, insulin and C-peptide levels 1 hour after BBR administration (Supplementary Table 2, shown below). We found no difference compared to placebo group. This suggests that BBR did not promote insulin secretion under fasting state in humans. Second, we have monitored ECG in participants and we have not found any changes between placebo and BBR groups (new Supplementary Fig. S12). Last, all participants were continually observed for 12 hours after the hyperglycemic clamp study to prevent hypoglycemia and we did not observe any delayed hypoglycemia or other side effects in BBR group.

Supplementary Table 2: Results from the hyperglycemic clamp study

	Placebo	BBR	P
Fasting blood glucose level-pretreatment (mmol/L)	4.87 ± 0.35	4.96 ± 0.27	NS
Fasting blood glucose level-posttreatment (mmol/L)	5.05 ± 0.22	5.11 ± 0.31	NS
Fasting plasma C-peptide level-posttreatment (ng/mL)	1.48 ± 0.29	1.65 ± 0.43	NS
Fasting plasma insulin level-posttreatment (μIU/mL)	5.39 ± 1.61	6.39 ± 2.11	NS
Steady-state blood glucose level (mmol/L)	11.92 ± 0.27	12.01 ± 0.29	NS
Coefficient of variation (%)	4.87 ± 1.34	5.57 ± 2.17	NS
GIR (mg/kg/min)	16.98 ± 2.21	17.54 ± 3.55	NS

Pretreatment and posttreatment fasting blood glucose levels indicate the blood glucose levels of volunteers before/after taking the placebo/BBR while not receiving the glucose injection. Data are presented as means ± SD.

We have added these points in the manuscript as followed (Page 13, Line 2-4): Notably, compared with the placebo, BBR did not change fasting blood glucose (FBG), fasting insulin or fasting proinsulin C-peptide levels (Supplementary Table S2). All subjects tolerated BBR well, and we observed no side effects. We also monitored ECG after the study and we did not observe difference between placebo and BBR groups (Supplementary Fig. S12).

b

N = 15	Placebo	BBR	P
HR (bpm)	66.8 ± 1.6	67.6 ± 1.7	0.757
PR (ms)	144.3 ± 3.5	147.9 ± 3.9	0.504
QRS (ms)	107.4 ± 1.9	106.4 ± 2.1	0.721
QT (ms)	385.8 ± 5.5	387.6 ± 5.6	0.820
QTc (ms)	406.0 ± 3.1	410.4 ± 3.5	0.374

Data are representative as mean ± SEM.

Supplementary Fig. 12: The ECG analysis from volunteers after the hyperglycemic study.

a, Representative ECG recorded from three volunteers receiving placebo or BBR. The volunteers were given placebo or BBR, and then performed the hyperglycemic clamp study, the ECG was performed after the hyperglycemic clamp. After the wash-out period, each volunteer received the other drug (BBR/placebo) and performed the ECG again.

b, ECG measurements of volunteers receiving placebo or BBR. Statistical significance was assessed using Student's *t* test (two-sided).

Reviewers' Comments:

Reviewer #3:

Remarks to the Author:

The author has addressed all comments satisfactorily and the paper is very well written.

REVIEWERS' COMMENTS

Reviewer #3 (Remarks to the Author):

The author has addressed all comments satisfactorily and the paper is very well written.

Answer: Thank you very much for your opinions.